# Brain-wide mapping of layer-specific functional connectivity in the human cortex at 3T using draining-vein-suppressed fMRI

**Wei-Tang Chang[1,2,3]\*, Weili Lin[1,2], Kelly S Giovanello[1,4]**

[1]Biomedical Research Imaging Center, University of North Carolina at Chapel Hill, Chapel Hill, United States; [2]Department of Radiology, University of North Carolina at Chapel Hill, Chapel Hill, United States; [3]Department of Biomedical Engineering, University of North Carolina at Chapel Hill, Chapel Hill, United States; [4]Department of Psychology and Neuroscience, University of North Carolina at Chapel Hill, Chapel Hill, United States

**\*For correspondence:**
weitang_chang@med.unc.edu

**Competing interest:** The authors declare that no competing interests exist.

## eLife Assessment

This **useful** study presents a possible solution for a significant problem - that of draining vein sensitivity in functional MRI, which complicates the interpretability of laminar-fMRI results. The addition of a low diffusion-weighted gradient is presented to remove the draining vein signal and obtain functional responses with higher spatial fidelity. However, the strength of the evidence is **incomplete**, and most tests appear to have been done only in a single subject. Significance thresholds in presented maps are very low and most cortical depth-dependent response profiles do not differ from baseline, even in the BOLD data shown as reference. Curiously, even BOLD group data fails to replicate the well-known pattern of draining towards the cortical surface.

**Abstract** Layer-dependent functional magnetic resonance imaging (fMRI) is a promising yet challenging approach for investigating layer-specific functional connectivity (FC). Achieving a brain-wide mapping of layer-specific FC requires several technical advancements, including sub-millimeter spatial resolution, sufficient temporal resolution, functional sensitivity, global brain coverage, and high spatial specificity. Although gradient echo (GE)-based echo planar imaging (EPI) is commonly used for rapid fMRI acquisition, it faces significant challenges due to the draining-vein contamination. In this study, we addressed these limitations by integrating velocity-nulling (VN) gradients into a GE-BOLD fMRI sequence to suppress vascular signals from the vessels with fast-flowing velocity. The extravascular contamination from pial veins was mitigated using a GE-EPI sequence at 3T rather than 7T, combined with phase regression methods. Additionally, we incorporated advanced techniques, including simultaneous multi-slice (SMS) acceleration and NOise Reduction with DIstribution Corrected principal component analysis (NORDIC PCA) denoising, to improve temporal resolution, spatial coverage, and signal sensitivity. This resulted in a VN fMRI sequence with 0.9 mm isotropic spatial resolution, a repetition time (TR) of 4 s, and brain-wide coverage. The VN gradient strength was determined based on results from a button-pressing task. Using resting-state data, we validated layer-specific FC through seed-based analyses, identifying distinct connectivity patterns in the superficial and deep layers of the primary motor cortex (M1), with significant inter-layer differences. Further analyses with a seed in the primary sensory cortex (S1) demonstrated the reliability of the method. Brain-wide layer-dependent FC analyses yielded results consistent with prior literature,

reinforcing the efficacy of VN fMRI in resolving layer-specific functional connectivity. Given the widespread availability of 3T scanners, this technical advancement has the potential for significant impact across multiple domains of neuroscience research.

## Introduction

Layer functional magnetic resonance imaging (layer fMRI) is an emerging field that measures the layer-specific activity noninvasively in humans. The ability to extract layer-specific signal provides an exciting opportunity to dissociate between bottom-up feedforward (FF) and modulatory feedback (FB) responses which were activated in separated layers of a cortical unit. However, implementing layer-dependent fMRI is technically challenging. First, obtaining layer-specific detail requires much higher spatial resolution than standard fMRI, with voxel sizes typically in the sub-millimeter range. This smaller voxel size inherently reduces the signal-to-noise ratio (SNR), which is why most layer-dependent fMRI studies rely on 7T scanners for enhanced sensitivity (*Brouwer et al., 2024*; *Dresbach et al., 2023*; *Lankinen et al., 2023*). Another challenge arises from the signal dependency across layers, particularly in blood oxygen level-dependent (BOLD) fMRI. BOLD contrast has been the gold standard for nearly three decades (*Bandettini et al., 1992*; *Ogawa et al., 1990*) due to its ability to acquire rapid images using gradient echo (GE)-based echo planar imaging (EPI; *Mansfield, 1977*). While arteries, capillaries, tissues, and draining veins all contribute to BOLD responses, the draining veins are considered the major contributor because of the larger volume and lower baseline oxygenation level compared to arteries (*Boas et al., 2008*). This reliance on draining veins presents a significant obstacle for layer-dependent fMRI using BOLD contrast. Intracortical veins, which run perpendicular or parallel to the cortical surface, drain into pial veins. As a result, BOLD signals originating in the lower cortical layers can be carried into the upper layers, a phenomenon known as the 'leakage model' (*Markuerkiaga et al., 2016*). Additionally, susceptibility changes near large pial vessels—often termed the 'blooming effect'—can lead to extravascular BOLD signal distortions in voxels distant from the vessel (*Kashyap et al., 2018*; *Li et al., 2012a*; *Moerel et al., 2018*).

In contrast to GE-EPI, spin-echo EPI (SE-EPI) primarily detects signals from the microvasculature, as contributions from large veins are largely suppressed. This suppression improves spatial specificity and minimizes cross-layer contamination caused by large vessels (*Boxerman et al., 1995*; *Duong et al., 2003*). However, SE-EPI generally exhibits lower functional sensitivity compared to GE-EPI because it refocuses BOLD dephasing, resulting in reduced overall signal changes during neural activation. Recently, a double spin-echo EPI method was introduced to enhance sensitivity in layer-dependent fMRI, achieving 0.8 mm isotropic resolution (*Han et al., 2021*). This approach demonstrated cortical activation peaking at approximately 1.0 mm from the surface of the primary motor cortex, indicating better layer specificity compared to GE-EPI. However, the inherently longer echo time (TE) of SE-EPI leads to a longer repetition time (TR) compared to GE-EPI. For the double spin-echo EPI method, the slab thickness is 12.8 mm with an effective TR of 6 s, making it challenging for brain-wide studies. Additionally, the extended EPI readout time required for submillimeter-resolution fMRI introduces T2* contamination, partially diminishing spatial specificity.

To address the issues of draining-vein contamination and blooming effect, cerebral blood volume (CBV)-based approaches have been gaining popularity in recent years. These methods are based on the principle that CBV changes primarily originate from small arterioles located near neural activation sites within specific layers (*Drew et al., 2011*; *Gagnon et al., 2015*). In contrast, larger downstream vessels are expected to have a limited contribution to overall CBV changes (*Drew et al., 2011*; *Gagnon et al., 2015*; *O'Herron et al., 2016*; *Takano et al., 2006*). Recently proposed CBV-based approaches include vascular space occupancy (VASO) fMRI (*Huber et al., 2017a*; *Lu et al., 2003*; *Lu et al., 2013*), integrated VASO and perfusion (VAPER; *Chai et al., 2020*), Magnetization transfer (MT) weighted laminar fMRI (*Pfaffenrot and Koopmans, 2022*), and Arterial Blood Contrast (ABC; *Priovoulos et al., 2023*). The conventional VASO technique takes advantage of the difference between the longitudinal relaxation times (T1) of tissue and blood, measuring the remaining extravascular water signal at the blood-nulling time point along blood T1 recovery after an inversion pulse. Upon neural activation, an increase in CBV leads to a relative increase in the amount of nulled blood volume within the voxel, causing a negative signal change. VAPER technique employs DANTE (Delay Alternating with Nutation for Tailored Excitation) preparation module (*Li et al., 2012b*) to suppress fast-moving spins such as

the vascular signal from arteries and veins but retain the stationary tissue signal. MT-weighted laminar fMRI enhanced the sensitivity to CBV by minimizing unwanted extravascular BOLD contributions from larger veins using MT preparation. Likewise, the ABC method enhances CBV weighting by selectively reducing venous and tissue signals through a pulsed saturation scheme.

Although CBV-based imaging approaches demonstrate high spatial specificity, the brain-wide acquisition time is relatively long due to the nature of the contrast-generation mechanism, which limits spatial coverage. Within the VASO fMRI framework, spatial coverage along slice direction is 18 mm and the acquisition time per slab is 3 s (*Huber et al., 2017a*). Leveraging the findings that CBV weighting can be achieved without imaging at strict blood-nulling time (*Ciris et al., 2014*; *Wu et al., 2008*), the MAGEC VASO technique (*Huber et al., 2021a*) further improved spatial coverage to 83.2 mm with 0.8 mm isotropic resolution, achieving a volumetric acquisition time of ~8 s, including both contrast preparation and signal readout. Other CBV-based approaches also require prolonged acquisition time for comparable spatial coverage. The VAPER method, for example, achieves an effective acquisition time of 12 s for 80.64 mm spatial coverage at 0.84 mm resolution (*Chai et al., 2024*). MT-weighted layer fMRI requires 40.1 s for a 24-mm-thick volume at 0.75 mm isotropic resolution, while the ABC method with 0.9 mm resolution covers a 27.9-mm-thick volume in 2.7 s. In addition to CBV-based approaches, cerebral blood flow (CBF)-based methods, such as perfusion imaging with zoomed arterial spin labeling (ASL), have demonstrated layer specificity at 7T (*Shao et al., 2021*). This method achieves a resolution of 1 mm isotropic with a slab thickness of 24 mm and an effective TR of 16.8 s. Among the CBV- and CBF-based techniques discussed above, the largest spatial coverage along the superior-inferior axis is 83.2 mm. Achieving brain-wide mapping of layer-specific functional connectivity, however, requires a field-of-view (FOV) size along the superior-inferior axis (z-axis) of around 120 mm to cover the entire cerebrum (*Mennes et al., 2014*). Assuming an inverse relationship between voxel size and acquisition time, we extrapolated acquisition times for the methods mentioned above based on a spatial coverage of 120 mm with a 0.9 mm isotropic resolution. Under these conditions, the volumetric acquisition times for MAGEC VASO, VAPER, MT-weighted, ABC, and ASL fMRI methods all exceed 9 s. Despite advancements in acquisition speed, current CBV/CBF-based fMRI techniques remain inadequate for layer-dependent resting-state fMRI. To investigate brain-wide mapping of resting-state functional connectivity, the Nyquist sampling rate per volume should be less than 5 s, as resting-state data is typically low-pass filtered at 0.1 Hz.

This study aims to reduce the inter-layer dependency while achieving whole-brain coverage with a volumetric repetition time (TR) under 5 s for layer-dependent fMRI. To accomplish this, we revisited the BOLD EPI method due to its rapid acquisition capabilities and explored approaches to reduce inter-layer dependency. Inter-layer dependency in BOLD signals arises from two distinct sources: extravascular and intravascular components. For the extravascular component, the blooming effect is a major source to inter-layer dependency. This effect primarily stems from large vessels, particularly pial veins, and can extend into remote tissue voxels. Consequently, depth-dependent profiles can be contaminated by these distant pial vessels. To mitigate the blooming effect while maintaining rapid acquisition, we employed a GE-EPI sequence at 3T instead of a 7T scanner. Although transitioning from 7T to 3T penalized BOLD sensitivity, it decreases the susceptibility-induced Larmor frequency shift by 57% and reduces extravascular signal by approximately 35% at 3T (*Uludağ et al., 2009*). The reduction in BOLD sensitivity can be partially compensated using NOise Reduction with DIstribution Corrected principal component analysis (NORDIC PCA; *Vizioli et al., 2021*), which has been shown effective in noise removal with minimal spatial blurring (*Dowdle et al., 2023*). Furthermore, we employed the phase regression method to alleviate the extravascular contamination from large pial veins (*Menon, 2002*).

For the intravascular component, addressing the draining-vein issue can involve estimating the depth-dependent BOLD signal through inverse calculations of the leakage model (*Havlicek and Uludağ, 2020*; *Heinzle et al., 2016*). However, the model-based vein removal methods are primarily exploratory, and the effectiveness is still in need of validation (*Huber et al., 2021a*). To enhance spatial specificity prospectively, small diffusion-weighted gradients, known as velocity-nulling (VN) gradients, can be incorporated into GE-EPI sequences. These gradients suppress signals from moving blood, thereby reducing contributions from pial and draining veins (*Boxerman et al., 1995*). Bipolar diffusion gradients have also been used in combination with other sequences, such as SE-BOLD and T2-preparation methods (*Duong et al., 2003*; *Huber et al., 2017a*). We selected GE-BOLD over

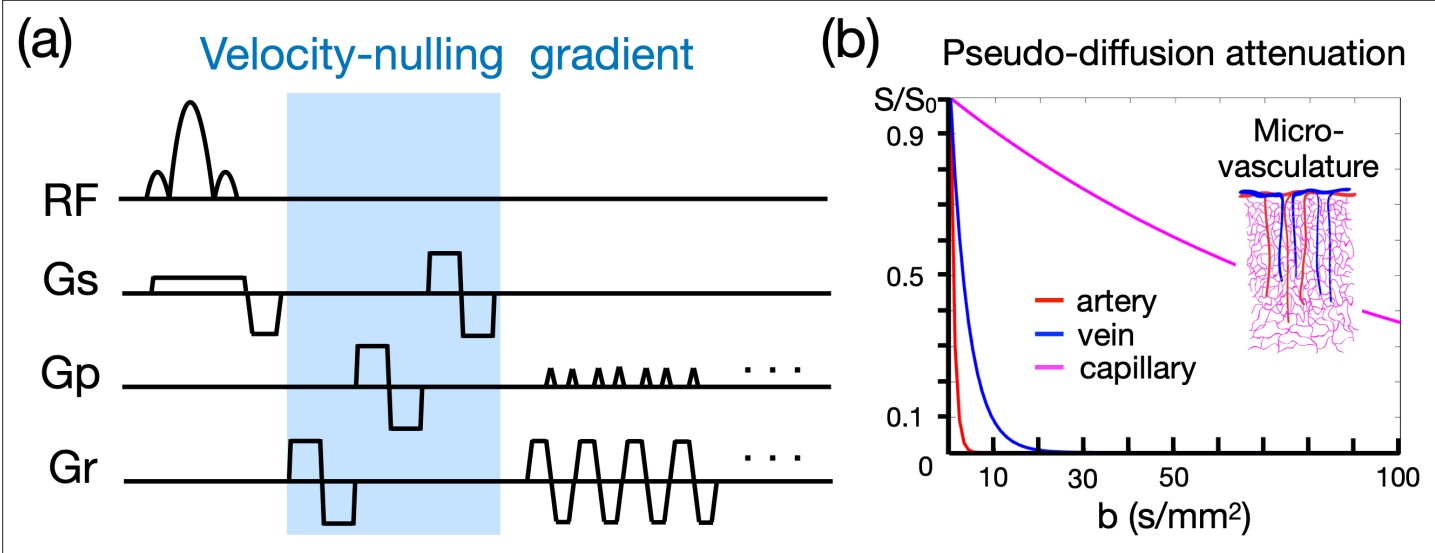

**Figure 1.** Velocity-nulling (VN) gradient in GE-EPI. (**a**) The diagram of pulse sequence. The VN gradient is highlighted in light blue. (**b**) The simulated signal attenuation against b values.

SE-BOLD sequences for its better sensitivity, as both SE-BOLD and diffusion-weighted T2-preparation sequences exhibit lower sensitivity. SE-BOLD sensitivity is approximately 20–30% lower than that of GE-BOLD, while diffusion-weighted T2-preparation sequences experience further sensitivity reduction due to incomplete refocusing (*Huber et al., 2017b*).

In this project, we implemented fMRI with 0.9 mm isotropic resolution, VN gradients, and brain-wide coverage at 3T. For volumetric acquisition, we applied the Simultaneous Multislice (SMS) method (*Setsompop et al., 2012*). This enabled whole-brain acquisition with 0.9 mm isotropic resolution, a FOV thickness of 113.4 mm along superior-inferior direction, and a TR of 4.0 s. To enhance sensitivity, we employed the NORDIC denoising method, while the phase regression approach was used to suppress macrovascular contributions from pial veins. The strength of the bipolar diffusion gradients, characterized by the b value, was optimized using a button-pressing task. Using the selected b value, we assessed the feasibility of brain-wide layer-dependent functional connectivity (FC) studies at 3T. This was achieved through layer-specific FC mapping with seeds placed in the primary motor cortex (M1) or the primary sensory cortex (S1). Additionally, we conducted a brain-wide layer-dependent connectome analysis using the Shen268 atlas (*Shen et al., 2013*). Our findings revealed multiple instances of layer-dependent functional connectivity that were consistent with previously reported results in the literature.

## Theory

### Draining-vein suppression using velocity-nulling gradient

The BOLD measurement at cortical layers is spatially blurred and biased towards superficial layers due to draining veins, including extravascular and intravascular components (*Duong et al., 2003*; *Turner, 2002*). Here, we incorporate a VN gradient into GE-EPI to suppress the fast-flowing intravascular signal from draining veins (see *Figure 1a*). The signal attenuation against the b value will generally follow the exponential decay in the Intravoxel Incoherent Motion (IVIM) process (*Le Bihan, 2019*). The IVIM process models the collective motion of water molecules in blood within a vessel network as they transition from one vessel segment to another. This collective movement can be perceived as a pseudo-diffusion process where average displacements equate to the mean vessel segment length and the mean velocity matches that of the blood in the vessels.

In a capillary network, the average displacement is approximately 100 µm and the mean velocity is around 1 mm/s, yielding a pseudo-diffusion coefficient (D*) of $10^{-8}$ m²/s (*Le Bihan, 2019*). In the case of cortical penetrating veins, the actual flow velocity is difficult to measure due to their small size. However, optical imaging suggests a mean flow velocity of ~2.5 mm/s in these veins (*Shih et al., 2013*). Assuming that inter-layer dependency arises primarily from the penetrating veins, we approximate an

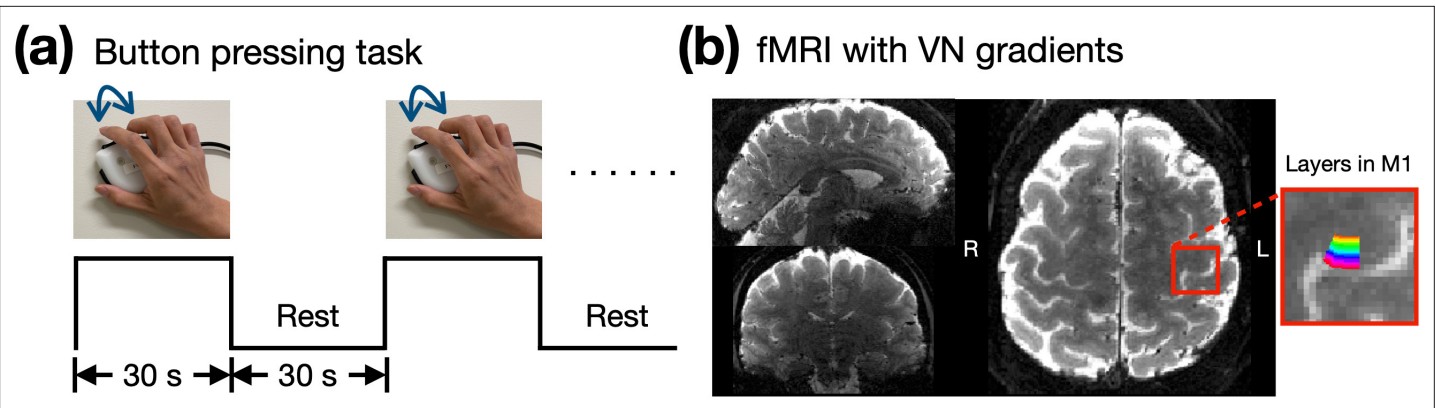

**Figure 2.** Empirical results of the button-press task showing varying levels of draining-vein suppression in different participants. (**a**) The paradigm of button-pressing task. (**b**) The acquired image with 0.9 mm isotropic resolution in three orthogonal views. The 20 layers of M1 are color-coded as shown in the right panel.

average displacement of 1 mm, based on cortical anatomical studies (*Reina-De La Torre et al., 1998*). Accordingly, the pseudo-diffusion coefficient for penetrating veins is ~$2.5 \times 10^{-7}$ m²/s. In contrast, the flow velocity in cortical arteries is ~12 mm/s (*Schaffer et al., 2006*), with an average displacement also of 1 mm. This configuration leads to a pseudo-diffusion coefficient of ~$1.2 \times 10^{-6}$ m²/s.

The signal amplitude that underlies the effect of the VN gradient follows an exponential decay as represented by the equation $S/S_0 = \exp[-b(D^* + D_{blood})]$. In this formula, S denotes the vascular signal affected by the VN gradient, $S_0$ denotes the signal strength without the VN gradient, b denotes the b value of diffusion gradient, $D^*$ indicates the pseudo diffusion coefficient, and $D_{blood}$ indicates the water diffusion coefficient in blood. As the $D^* \gg D_{blood}$, the signal decay equation can be simplified to $S/S_0 = \exp(-b \bullet D^*)$. Using the predetermined pseudo diffusion coefficients for arteries, veins, and capillaries, we simulated the signal attenuations as illustrated in *Figure 1b*. Our results imply that a relatively small b value should effectively suppress the contribution from draining veins, yet simultaneously cause only minimal attenuation of signals originating from capillaries.

## Results

### Optimization of VN gradient strength

We evaluated the effects of the VN gradient at different b values by examining the depth-dependent BOLD activation profile in the primary motor cortex (M1) during a button-pressing task as shown in *Figure 2a*. *Figure 2b* displays the GE-EPI image in three orthogonal views, highlighting the M1 region within a red box, with a zoom-in view and a color-coded layer representation on the right.

The NORDIC denoising method was applied to enhance sensitivity. Recent studies have demonstrated that the NORDIC method effectively removes noise while introducing minimal spatial blurring (*Dowdle et al., 2023*). To confirm this in our data, *Figure 3a and b* compare the BOLD activation maps before and after NORDIC denoising in the visual and motor cortices, respectively. These maps correspond to a TE of 30ms without phase regression, ensuring the effects of VN gradients and phase regression were controlled. The activation patterns in the denoised maps were consistent with those in the non-denoised maps but showed higher statistical significance. Notably, BOLD activation within M1 was only observed after NORDIC denoising, highlighting the necessity of applying this approach. *Figure 3c* presents the depth-dependent activation profiles in M1, highlighted by the green contours in *Figure 3b*. Both profiles exhibited similar trends. However, the non-denoised profile showed larger confidence intervals compared to the NORDIC-denoised profile, as expected. These results confirm that NORDIC denoising enhances sensitivity without distorting the functional signal.

Depth-dependent BOLD activation profiles in M1 were extracted using a voxel-based approach. *Figure 4a–d* illustrates these profiles across different subjects, with each column representing different TEs and b values. Three out of four subjects exhibited double-peak response patterns. Notably, using a b value of 8 s/mm² appeared to excessively suppress deep-layer activation. Comparing column 2nd and 3rd (b=0 and 6 s/mm², respectively, at TE = 38 ms) showed that the percentage of BOLD signal

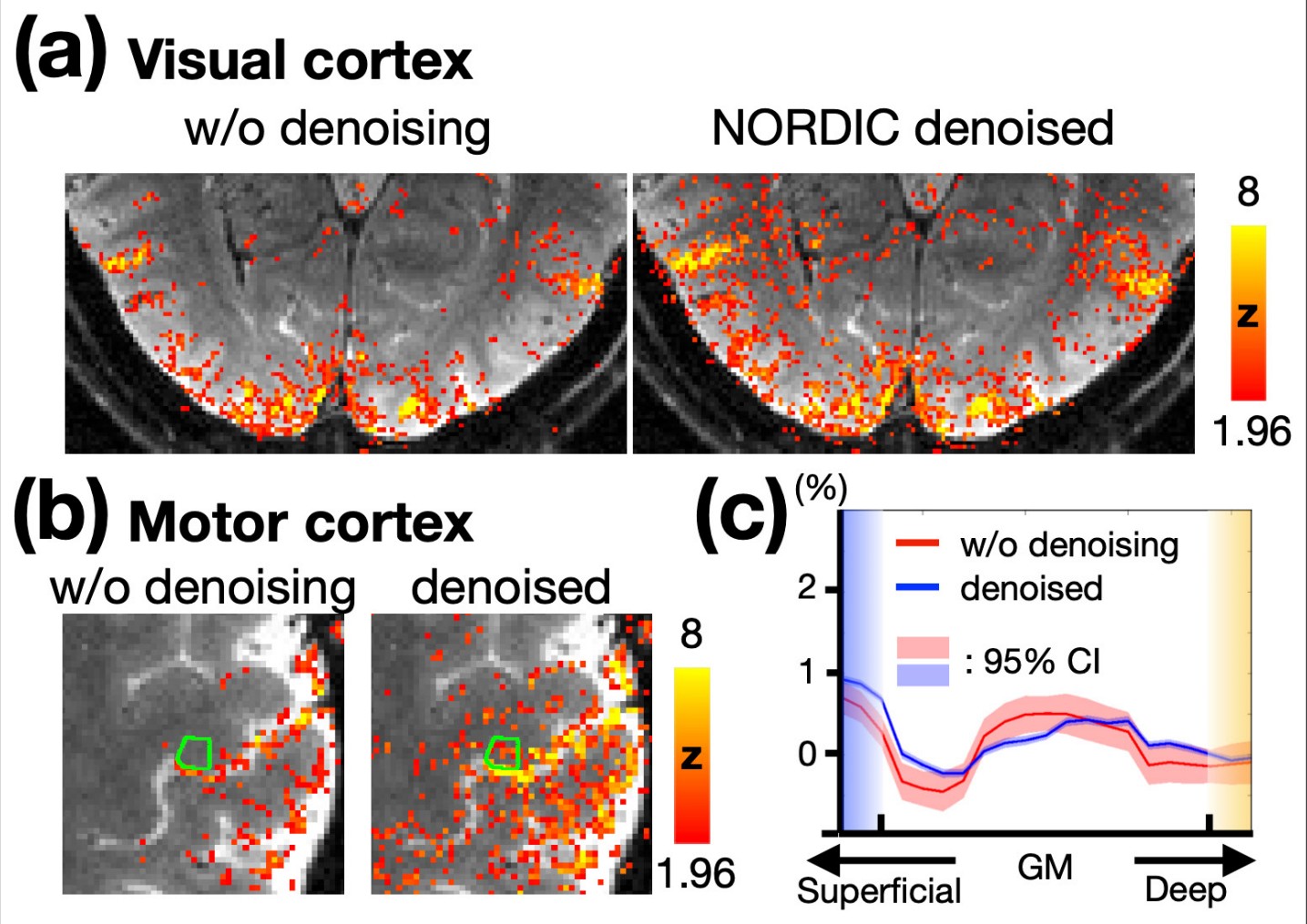

**Figure 3.** Effects of NORDIC denoising on BOLD activation profiles. (**a**) BOLD activation maps in the visual cortex without and with NORDIC denoising. (**b**) BOLD activation maps in the motor cortex without and with NORDIC denoising. The statistical maps were corrected (uncorrected *P*<0.05; corrected *P*<0.05) and color-coded as indicated by the color bar. (**c**) Depth-dependent BOLD activation profiles for the M1 regions shown in (**b**). The shaded areas in light red and blue represent the 95% confidence intervals.

change in superficial layers is generally lower with b of 6 s/mm² than with b of 0, suggesting that VN gradient-induced signal suppression is more pronounced in superficial layers. Moreover, *Figure 4a–d* also presents layer-dependent profiles with and without phase-regression (PR). Observable differences were noted in only two subjects, as indicated by the yellow arrows in the first column of *Figure 4c* and the second column of *Figure 4d*. Overall, PR primarily influenced the superficial layers within M1, though its impact was not substantial.

To investigate whether the low efficacy of phase regression observed in a small M1 region could be generalized to other brain regions, we evaluated its efficacy in the visual cortex using the same datasets of subject #1. All four parameter combinations were tested. The columns in *Figure 5* represent activation maps under different parameter combinations, while the rows show the maps without phase regression, the maps with phase regression, and the difference maps, from top to bottom. The difference maps demonstrate that phase regression is effective across all tested conditions, at least when TE ≤39ms and b≤8 s/mm². Consistent with the findings in *Figure 4*, the reduction of macrovascular-induced effects was most prominent near the pial surface. As highlighted by the blue circles in *Figure 5*, VN gradients effectively suppressed macrovascular signals as well. Furthermore, unlike the results in *Figure 4*, the reduction in activation strength in the visual cortex was less obvious when using a b value of 8 s/mm². This suggests that the degree of signal reduction associated with

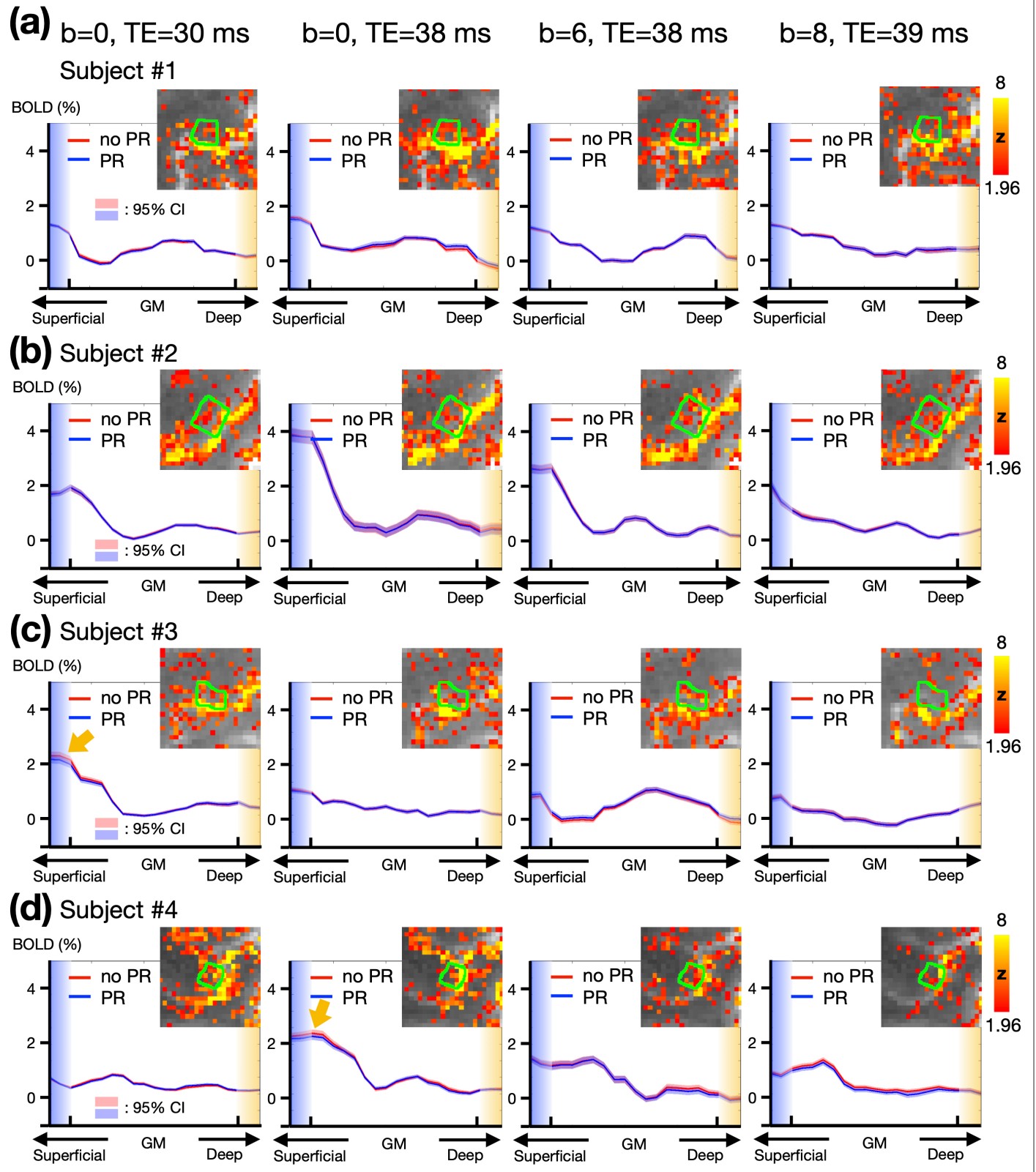

**Figure 4.** Empirical results of the button-press task showing varying levels of draining-vein suppression in different participants. (**a–d**) Depth-dependent BOLD activation profiles in M1. The green contours represent the edges of M1 regions. Each row represents data from a different individual. Columns from left to right correspond to TE = 30 ms, TE = 38 ms, TE = 38 ms with b=6 s/mm², and TE = 39 ms with b=8 s/mm². The statistical maps were corrected (uncorrected p<0.05; corrected p<0.05) and color-coded as indicated by the color bar. The shaded areas in light red and blue represent the 95% confidence intervals (CI).

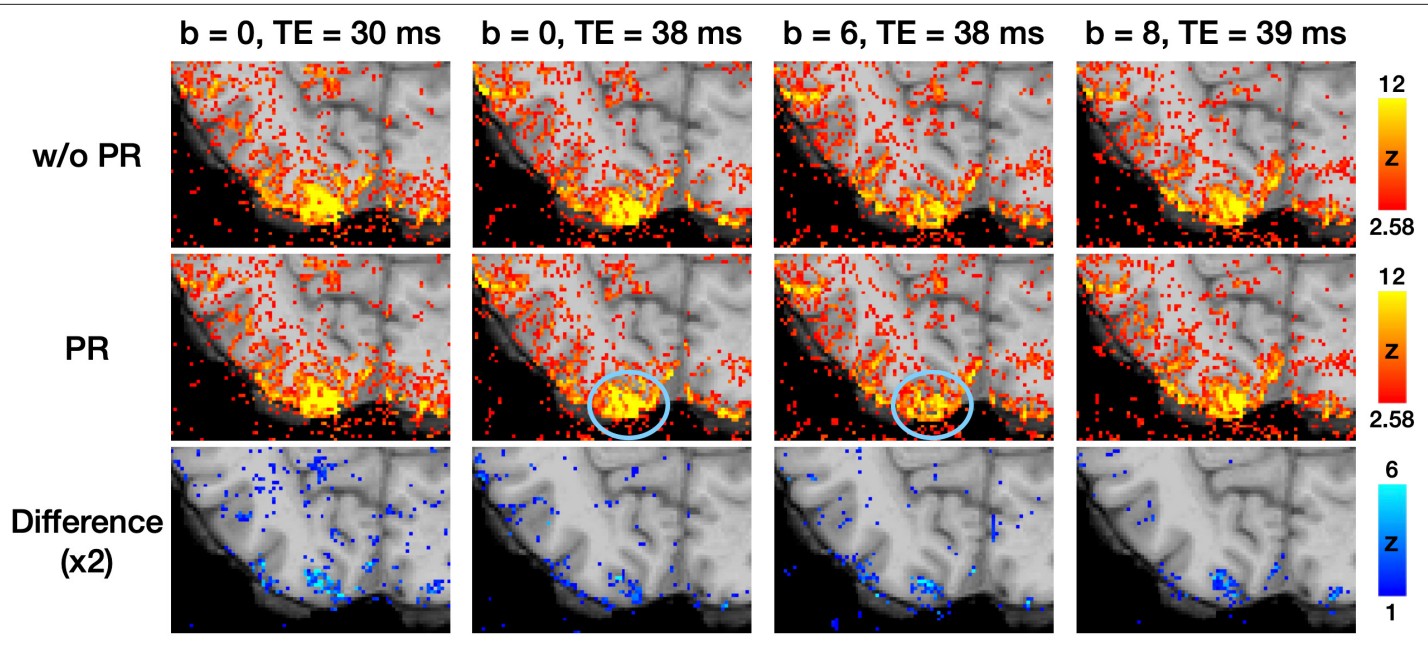

**Figure 5.** The activation maps correspond to visual cues from the button-pressing task. The difference maps were generated by subtracting the activation maps in the middle row (with phase regression) from those in the top row (without phase regression). The blue circles highlight the effective suppression of macrovascular signals by VN gradients.

b values may vary across brain regions. Based on the findings from *Figures 4 and 5*, we selected a b value of 7 s/mm² as a reasonable compromise and employed it in subsequent experiments.

## Assessment of inter-layer dependency

To investigate the effects of VN gradients on signal dependency across cortical depth, we resampled the fMRI time series, both with and without VN gradients, onto the brain surface at different cortical depths. For each cortical vertex, intracortical FC matrices were computed and subsequently averaged across the entire brain surface. As shown in *Figure 6a*, the reduction in layer dependency is most pronounced in the superficial layers. Furthermore, reductions in layer dependency were observed between the superficial and middle layers, as well as between the middle and deep layers, with these regions highlighted by the red circle. These reductions induced by VN gradients were statistically

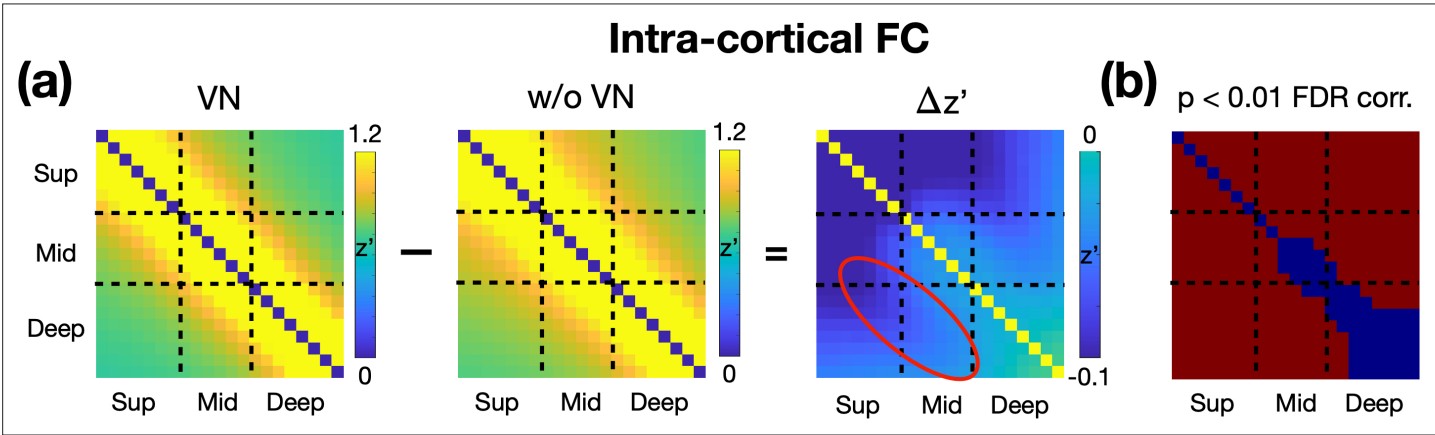

**Figure 6.** Impact of VN gradients on intracortical FC matrices. (**a**) The panels from left to right display the intracortical FC matrix derived from VN fMRI, the intracortical FC matrix derived from regular fMRI, and the difference between the two matrices. All FC matrices are Fisher's z-transformed. (**b**) Statistical significance of the differences in the FC matrices. Red regions indicate the differences are statistically significant (FDR-corrected p-value <0.05).

significant (FDR-corrected p<0.05), as illustrated in *Figure 6b*. The observed reductions in layer-dependent FC argue against the possibility that these changes are merely due to layer-independent signal attenuation. If VN gradients attenuated the signal without suppressing cross-layer blurring, the result would be a more uniform decrease in FC across all layers. Instead, the findings suggest that VN gradients selectively influence intracortical FC, supporting their role in improving layer-dependent specificity.

## Functional connectivity maps using layer-specific seeds in M1

The sensorimotor network has been studied extensively, not only region-specific but also layer-specific characteristics on functional connectivity (*Cauller, 1995*; *Felleman and Van Essen, 1991*). In the primary motor cortex, superficial layers are predominantly associated with sensory areas, while deep layers are primarily linked to premotor areas. To evaluate these layer-specific properties in our resting-state data, we conducted the layer-specific FC analysis as illustrated in *Figure 7a*. Superficial and deep layer labels for M1 were generated in EPI space, with volume-based labels derived from individual surface-based M1 labels. The seed time courses were extracted from the superficial and deep layers in M1, and these time courses were used as regressors in the GLM analysis across the brain. The resulting FC maps were resampled onto individual cortical surfaces at various depths and subsequently morphed to a template cortical surface for group-level analysis (refer to the Materials and methods section for further details).

*Figure 7b* shows a comparison of FC maps derived from superficial-layer and deep-layer seeds. To evaluate the layer specificity of these FC patterns, *Figure 7c* displays the magnitude of cross-layer differences based on the FC maps shown in *Figure 7b* In *Figure 7b*, the first row illustrates overall FC across cortical depth, with conventional fMRI results (without VN gradients) shown in the left panel and VN fMRI results in the right panel. The second through fourth rows highlight FC maps for superficial, middle, and deep layers, respectively. Generally, deep M1 exhibited stronger FC compared to superficial M1 within the sensorimotor network and beyond. In addition, the FC patterns obtained from conventional fMRI appeared more spatially diffuse compared to those from VN fMRI, reflecting the latter's improved specificity. The VN fMRI results further revealed that superficial M1 exhibited stronger connectivity with superficial S1, particularly Brodmann area (BA) 1, as shown in the superficial-layer FC maps. This observation aligns with previous reports of connectivity between superficial M1 and sensory regions (*Cauller, 1995*). The findings in *Figure 7c* further indicate that superficial M1 primarily connects with superficial layers of S1, with relatively weaker connectivity to middle or deep layers.

*Figure 7c* examines the inter-layer dependency. If cross-layer signals are blurred, such as by contamination from draining veins or the blooming effect, the statistical distinctions across layers in FC maps would diminish. In *Figure 7c*, conventional fMRI demonstrated cross-layer differences only in the M1 and S1 regions. In contrast, VN-enhanced fMRI showed a significantly greater number of regions exhibiting cross-layer differences. For example, as highlighted by the green circles in *Figure 7b*, both conventional and VN-enhanced fMRI showed stronger FC between the angular gyrus (AG) and deep M1 compared to superficial M1. However, while conventional fMRI failed to detect significant differences between superficial and middle layers, VN fMRI identified these differences as statistically significant, as highlighted by green circles in *Figure 7c*.

Furthermore, the deep-layer seed in M1 is expected to exhibit stronger FC with deep layers than with middle layers within the same region. For VN fMRI, this pattern of stronger deep-layer connectivity was observed across nearly the entire M1 in *Figure 7c*, whereas it was restricted to less than half of the M1 in conventional fMRI, as shown by the blue circles. Additionally, the brown circles in *Figure 7b* indicate that deep layers in Wernicke's area exhibited FC with deep M1 in both conventional and VN fMRI. However, as shown by the brown circles in *Figure 7c*, statistical analyses revealed that this connectivity was preferentially associated with the deep layers rather than the superficial layers in VN fMRI. Such a layer-specific distinction was not evident in conventional fMRI, highlighting the enhanced layer specificity of VN fMRI.

To evaluate whether the spatially diffuse FC associated with deep-layer M1 arises from artifacts specifically related to deep-layer signals, we conducted similar FC analyses using a seed region in S1. From a neurophysiological perspective, the S1 region primarily supports bottom-up processing and is expected to exhibit FC that is more confined to the sensorimotor network. If the patterns observed in

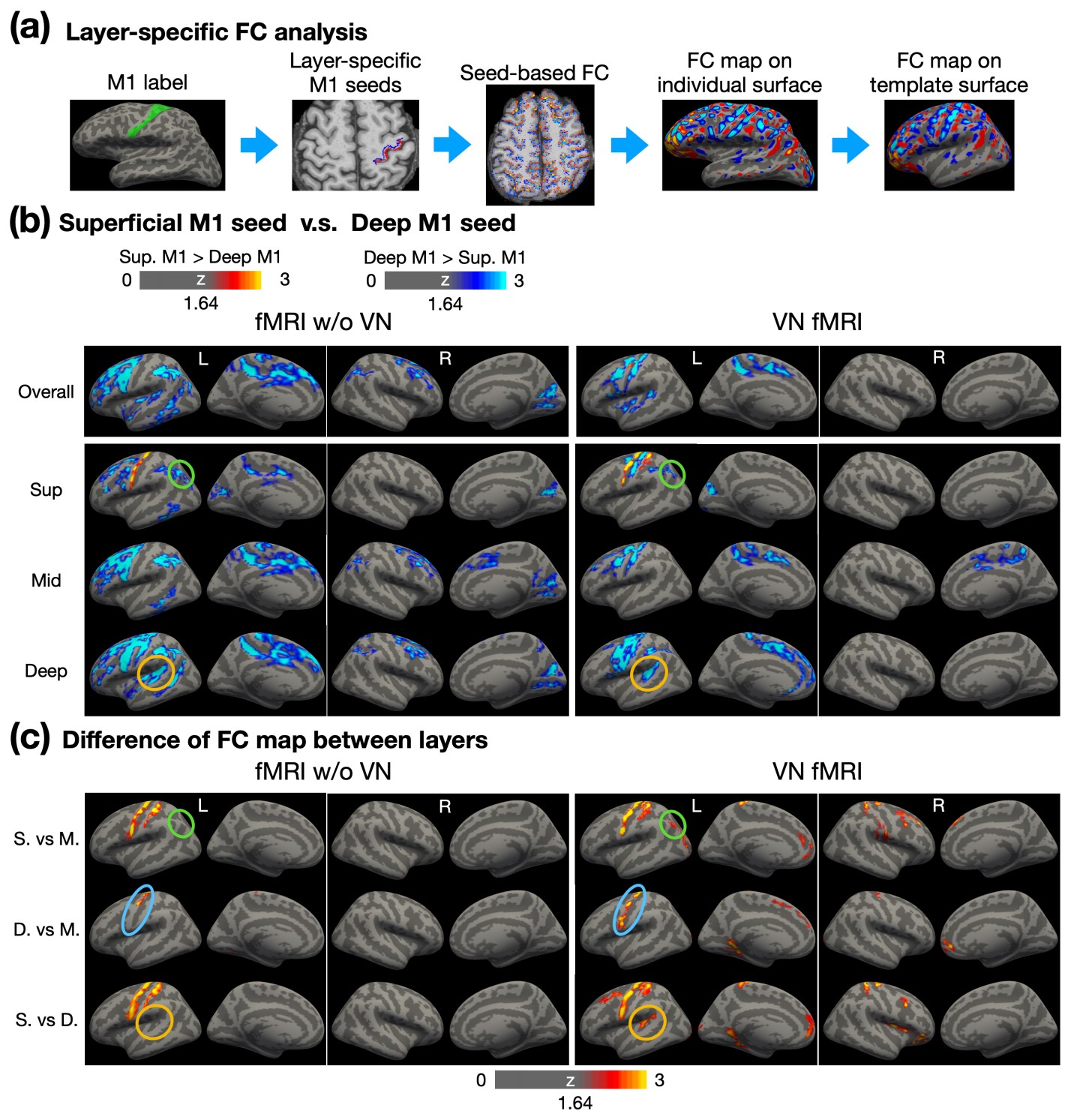

**Figure 7.** Layer-dependent FC analysis using layer-specific M1 seeds. (**a**) Workflow for conducting layer-specific FC analysis. Surface-based labels were converted from individual surface to volume space. Red and blue colors represent the superficial and deep layers, respectively. Seed-based FC analysis was performed in EPI space, followed by morphing to individual surfaces and then to the template surface. (**b**) Contrast of FC maps corresponding to superficial, middle, and deep layers. Warm colors represent stronger FC associated with superficial layers in M1, while cool colors indicate stronger FC associated with deep layers in M1. (**c**) Differential FC maps showing the magnitude of differences between layers, including superficial vs. middle, deep vs. middle, and superficial vs. deep layers. Abbreviations: Sup – superficial; Mid – middle; S. – superficial; M. – middle; D. – deep.

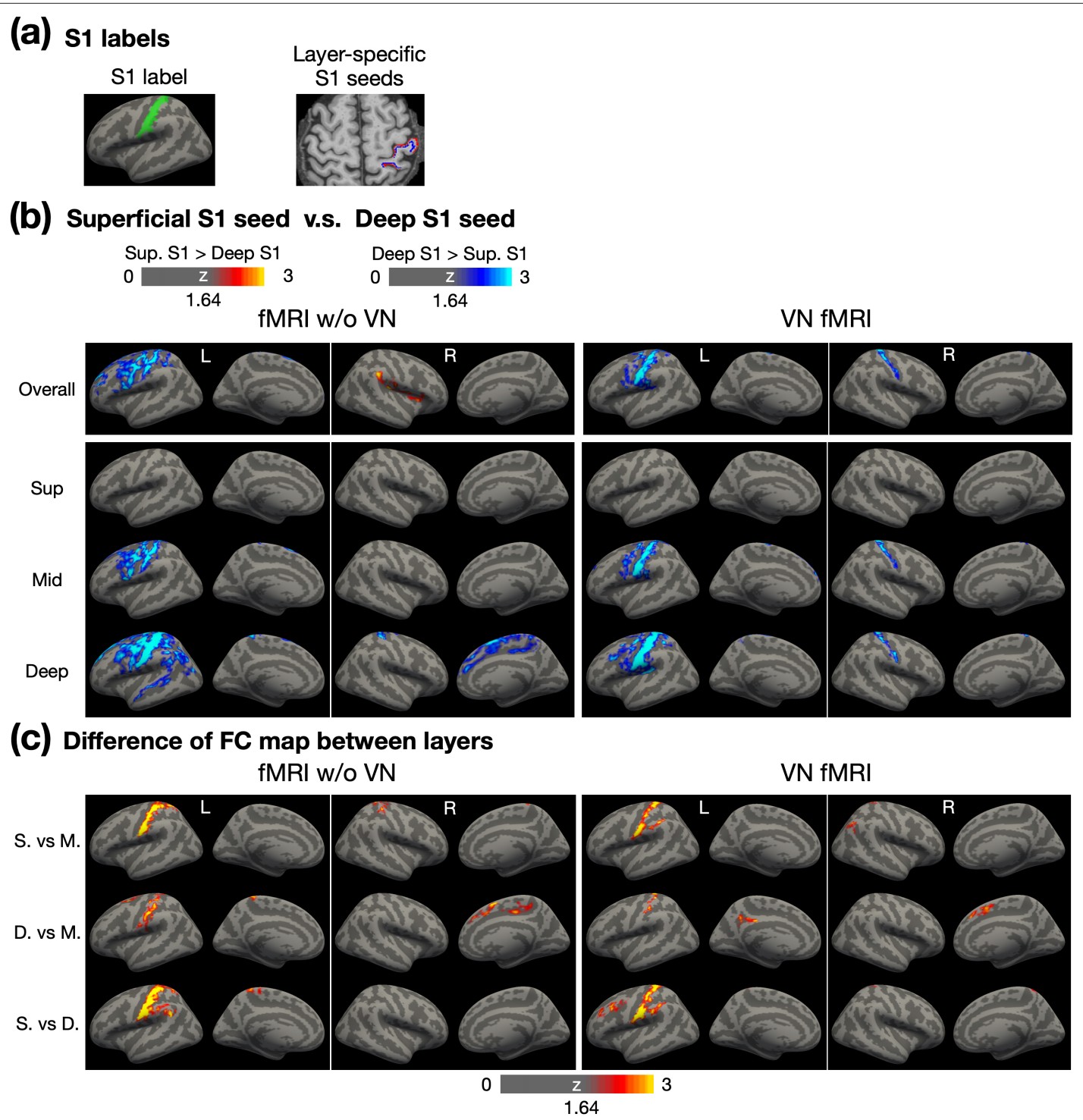

**Figure 8.** Layer-dependent FC analysis using layer-specific S1 seeds. (**a**) The illustration of surface-based and volume-based S1 labels. (**b**) Contrast of FC maps corresponding to superficial, middle, and deep layers. Warm colors represent stronger FC associated with superficial layers in S1, while cool colors indicate stronger FC associated with deep layers in S1. (**c**) Differential FC maps showing the magnitude of differences between layers, including superficial vs. middle, deep vs. middle, and superficial vs. deep layers. Abbreviations: Sup – superficial; Mid – middle; S. – superficial; M. – middle; D. – deep.

*Figure 7* were driven by deep-layer-related artifacts, the results for S1 would be expected to demonstrate diffuse FC patterns similarly, given the close anatomical proximity of M1 and S1.

*Figure 8a* illustrates the S1 seed placement. The left panel shows the surface-based S1 label on the template surface, while the right panel displays superficial and deep-layer S1 labels on an individual brain. Following a similar layout to *Figure 7b*, the top to bottom rows in *Figure 8b* represent the FC maps for overall, superficial, middle, and deep layers, respectively. The FC maps of conventional fMRI appeared to be more spatially diffuse, extending beyond the sensorimotor network. In contrast, the FC maps of VN fMRI are relatively confined to the sensorimotor network, consistent with neurophysiological expectations. These findings suggest that the stronger and more widespread FC associated with deep-layer M1 was unlikely due to deep-layer-related artifacts.

In the FC maps of VN fMRI corresponding to the superficial layer (*Figure 8b*), no significant differences were observed between superficial- and deep-layer FC maps, indicating that the superficial and deep layers in S1 exhibited comparable FC strength across the cerebral cortex. However, in the middle- and deep-layer FC maps, the deep-layer S1 demonstrated stronger FC than the superficial-layer S1, particularly in bilateral S1 and the left S1 and M1 regions. Cross-layer statistical maps in *Figure 8c* further revealed that the deep-layer seed in S1 predominantly connected with the middle

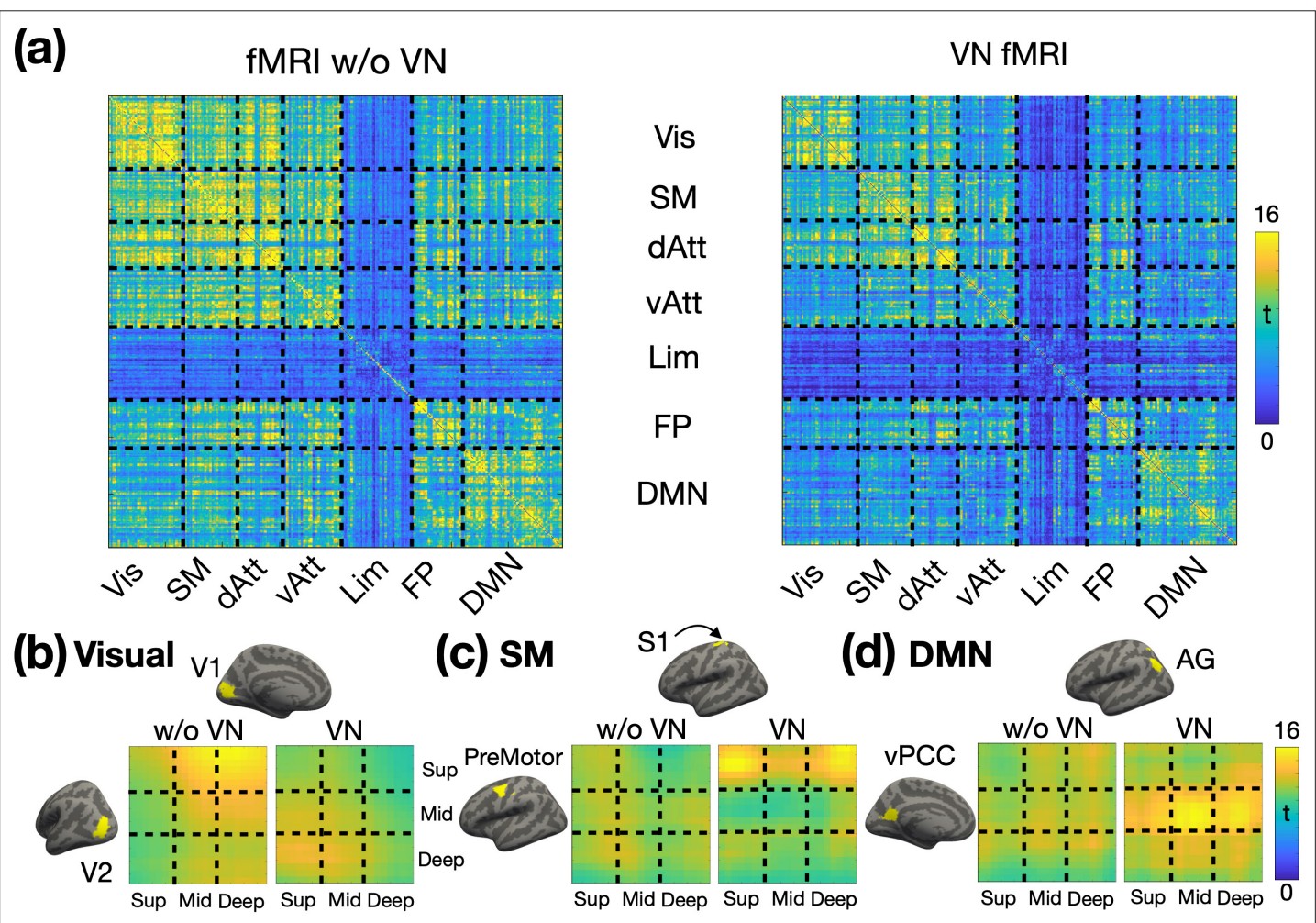

**Figure 9.** Layer-specific functional connectivity across functional networks (N=14). (**a**) Fisher's z transformed FC matrices. The FC matrix on the left represents the Fisher's z values without a VN gradient, while the matrix on the right corresponds to the Fisher's z values with a VN gradient. From (**b**) to (**d**) displays the depth-dependent FC matrices for a representative ROI pair in visual, sensorimotor, and default-mode networks, respectively. T-values are color-coded as indicated by the color bar. The left panel displays the matrix without a VN gradient, and the right panel presents the matrix with a VN gradient. Abbreviations: V1 – primary visual cortex; V2 – secondary visual cortex; S1 – primary sensory cortex; vPCC – ventral posterior cingulate cortex; AG – angular gyrus; Vis – visual network; SM – sensorimotor network; dAtt – dorsal attention network; vAtt – ventral attention network; Lim – limbic network; FP – frontoparietal network; DMN – default-mode network.

and deep layers within S1, rather than with the superficial layers. Notably, other regions showing significantly stronger FC associated with the deep-layer S1 did not exhibit clear layer specificity, potentially due to an insufficient sample size.

## Brain-wide layer-specific functional connectivity

To explore brain-wide layer-specific functional connectivity, we parcellated the brain using Shen268 functional atlas (refer to the Materials and methods section for further details). The group-level statistics for the layer-specific functional connectivity matrices are presented in *Figure 9a*, with ROIs organized according to their respective functional networks. The FC matrix on the left was derived from conventional fMRI data without the application of VN gradients, while the matrix on the right was generated using VN fMRI data. In general, the FC matrix from conventional fMRI displayed more widespread connectivity than that of VN fMRI, consistent with the seed-based FC results shown in *Figures 7 and 8*.

The enhanced layer specificity provided by VN fMRI enables detailed mapping of the layer-specific brain connectome. Examples from the visual, sensorimotor, and default-mode functional networks are presented in *Figures 7c and d and 9b*, respectively. The surface-based labels of ROI pairs are displayed on the left and upper sides of the depth-dependent FC matrices. Focusing on VN fMRI results, within the visual network, the primary visual cortex (V1) exhibited FB connectivity from the deep layers of the visual area V2 to the superficial layers of V1. In the sensorimotor network, the premotor area received inputs exclusively in its superficial layers, originating from both the superficial and deep layers of the S1. Within the default mode network (DMN), the ventral posterior cingulate cortex (vPCC) demonstrated connectivity dominated by its middle layers. Notably, the angular gyrus (AG) exhibited FF connectivity to the middle layers of vPCC, consistent with its role as a major hub in the DMN and in alignment with previously reported findings from a laminar-specific fMRI study. In contrast, the results from conventional fMRI showed diffuse and less distinct patterns in the depth-dependent FC matrices, diverging from established findings in the literature. This further underscores the capability of VN fMRI in resolving layer-specific connectivity with greater accuracy.

## Discussion

To develop brain-wide layer-specific functional connectivity using depth-dependent functional information, several technical criteria must be met: sub-mm spatial resolution, high spatial specificity, adequate temporal resolution, sufficient functional sensitivity, and global brain coverage. In this study, we addressed those challenges by incorporating a number of techniques, such as velocity-nulling gradients, SMS acceleration, phase regression, and NORDIC denoising techniques. As a result, we were able to develop a VN fMRI sequence featuring 0.9 mm isotropic spatial resolution, a TR of 4 s and brain-wide coverage. The effects of TE and b values associated with the VN gradient were evaluated. Based on fMRI results from a button-pressing task, a b value of 7 s/mm$^2$ was selected to achieve a reasonable balance between sensitivity and specificity. Using this parameter, we assessed the validity of layer-specific FC through seed-based analyses. Results revealed distinct FC patterns associated with superficial and deep layers in M1, and the resulting FC maps exhibited significant differences between layers, demonstrating that VN fMRI enhances inter-layer independence. Additional FC analyses were also conducted with a seed placed in S1 to further validate the findings. Moreover, brain-wide layer-dependent FC analyses revealed several findings consistent with existing literature, supporting the efficacy and reliability of the VN fMRI approach in resolving layer-dependent functional connectivity.

The proposed VN-fMRI method employs VN gradients to selectively suppress signals from fast-flowing blood in large vessels. Although this approach may initially appear to diverge from the principles of CBV-based techniques (*Chai et al., 2020*; *Huber et al., 2017a*; *Pfaffenrot and Koopmans, 2022*; *Priovoulos et al., 2023*), which enhance sensitivity to vascular changes in arterioles, capillaries, and venules while attenuating signals from static tissue and large veins, it aligns with the fundamental objective of all layer-specific fMRI methods. Specifically, these approaches aim to maximize spatial specificity by preserving signals proximal to neural activation sites and minimizing contributions from distal sources, irrespective of whether the signals are intra- or extravascular in origin. In the context of intravascular signals, CBV-based methods preferentially enhance sensitivity to functional changes

in small vessels (proximal components) while demonstrating reduced sensitivity to functional changes in large vessels (distal components). For extravascular signals, functional changes are a mixture of proximal and distal influences. While tissue oxygenation near neural activation sites represents a proximal contribution, extravascular signal contamination from large pial veins reflects distal effects that are spatially remote from the site of neuronal activity. CBV-based techniques mitigate this challenge by unselectively suppressing signals from static tissues, thereby highlighting contributions from small vessels. In contrast, the VN fMRI method employs a targeted suppression strategy, selectively attenuating signals from large vessels (distal components) while preserving those from small vessels (proximal components). Furthermore, the use of a 3T scanner and the inclusion of phase regression in the VN approach mitigates contamination from large pial veins (distal components) while preserving signals reflecting local tissue oxygenation (proximal components). By integrating these mechanisms, VN fMRI improves spatial specificity, minimizing both intravascular and extravascular contributions that are distal to neuronal activation sites.

Bipolar diffusion gradients have been employed to nullify signals from fast-flowing blood, as demonstrated by *Boxerman et al., 1995*. Their work showed that vessels with flow velocities producing phase changes greater than π radians due to the bipolar gradients experience significant signal attenuation. The critical velocity for such attenuation can be calculated using the formula: $1/(2\gamma G\Delta\delta)$ where $\gamma$ is the gyromagnetic ratio, G is the gradient strength, $\delta$ is the gradient pulse width, and $\Delta$ is the time between the two bipolar gradient pulses. In the framework of Boxerman et al. at 1.5T, the critical velocity for b value of 10 s/mm² is ~8 mm/s, resulting in a~30% reduction in functional signal. In our 3T study, b values of 6, 7, and 8 s/mm² correspond to critical velocities of 16.8, 15.2, and 13.9 mm/s, respectively. The flow velocities in capillaries and most venules remain well below these thresholds. Notably, in our VN fMRI sequences, bipolar gradients were applied in all three orthogonal directions, whereas in Boxerman et al.'s study, the gradients were applied only in the z-direction. Given the voxel dimensions of $3 \times 3 \times 7$ mm³ in the 1.5T study, vessels within a large voxel are likely oriented in multiple directions, meaning that only a subset of fast-flowing signals would be attenuated. Therefore, our approach is expected to induce greater signal reduction, even at the same b values as those used in Boxerman et al.'s study.

Bipolar diffusion gradients have also been employed in layer-dependent fMRI at 7T, but in conjunction with T2 preparation methods (*Huber et al., 2017a*) rather than rapid GE-EPI sequences. This is probably due to the limited TE associated with GE-EPI at 7T, which is insufficient to accommodate bipolar diffusion gradients in an ultra-high-resolution regime. However, it is noteworthy that the functional sensitivity of diffusion-weighted T2 preparation methods is considerably lower than the GE-BOLD method (*Huber et al., 2021a*). While higher magnetic field strengths could enhance sensitivity, the extravascular effect in GE-BOLD method decreases the spatial specificity. The implementation of VN fMRI at 3T not only addresses the extravascular effect but also leverages several unique advantages of 3T over 7T. First, the extended T2* values at 3T lead to diminished T2* blurring along the phase-encoding direction, thus yielding a more focused point-spread function (PSF). Second, geometric distortions and signal losses due to susceptibility effects are less prominent at 3T relative to those at 7T. Third, the feasibility of conducting layer-dependent fMRI at 3T broadens the scope for multi-site, large-scale investigations. These attributes are of critical importance for a broad range of neuroscience applications.

In addition to draining-vein contamination, another source of cross-layer blurring is extravascular effect, which commonly originates from large pial veins. According to the biophysical models (*Ogawa et al., 1993*), the extravascular contamination from the pial surface is inversely proportional to the square of the distance from vessel. For a vessel diameter of 0.3 mm and an isotropic voxel size of 0.9 mm, the induced frequency shift is reduced by at least 36-fold at the next voxel. Notably, a vessel diameter of 0.3 mm is larger than most pial vessels. Theoretically, the extravascular effect contributes minimally to inter-layer dependency, particularly at 3T compared to 7T due to weaker susceptibility-related effects at lower field strengths. Empirically, as shown in *Figure 7c*, the results at M1 demonstrated that layer specificity can be achieved statistically with the application of VN gradients.

Although extravascular effects have minimal impact on signals from the middle and deep cortical layers (*Markuerkiaga et al., 2016*), signal contamination from pial veins can still affect the superficial layers. This study addressed this issue by employing phase regression, a method used to reduce biases toward signals from superficial cortical layers (*Knudsen et al., 2023*; *Koopmans et al., 2010*;

Markuerkiaga et al., 2021; Scheeringa et al., 2016). Similarly, our findings in Figures 4 and 5 demonstrate that phase regression effectively suppresses macrovascular contributions primarily near the gray matter/CSF boundary, regardless of whether VN gradients are applied. These results suggest that phase regression can complement VN gradients, offering a robust strategy to enhance spatial specificity in layer-specific fMRI.

The distinction between FC maps associated with superficial- and deep-layer seeds was used to evaluate the reliability of layer-dependent resting-state FC. In the work of Huber et al., 2017a, the superficial- and deep-layer seeds at M1 were defined manually and the corresponding FC maps were restricted to the same 2D image plane as the seeds. Their results showed that superficial M1 exhibited FC with a region in the post-central gyrus, while deep-layer M1 connected with a region in the pre-central gyrus. In our study, superficial- and deep-layer seeds in M1 were generated automatically using predefined anatomical labels from FreeSurfer. The resulting FC maps were consistent with previous findings, showing that superficial-layer M1 functionally connected with post-central regions, while deep-layer M1 exhibited FC with pre-central regions. Additionally, our results revealed that deep-layer M1 was functionally connected to the deep layers of Wernicke's area. As part of the language network (Tomasi and Volkow, 2012) and primarily involved in language comprehension (Johns, 2014), Wernicke's area likely integrates information through long-range corticocortical projections (Huber et al., 2021a; Lawrence et al., 2019). Moreover, we observed that deep-layer M1 connected with the superficial layer of the dorsal angular gyrus (AG), a critical hub in the default-mode network (Biswal et al., 1995; Biswal, 2012). This connectivity suggests that M1 and AG are not functionally isolated even during rest. The deep-to-superficial connectivity implies that motor systems may provide contextual signals to high-order cognitive regions, potentially via a FB pathway. These findings suggested the integrative role of M1 in facilitating communication across functional networks during resting state.

The analysis of brain-wide layer-dependent FC is a cutting-edge approach that requires sufficient spatial specificity, brain coverage, sensitivity, and sampling rates to be feasible. Some of the previously mentioned methods have investigated layer-dependent FC within specific networks, such as the visual network (Koiso et al., 2022) or the cortico-thalamic network (Deshpande et al., 2022). The MAGEC VASO method (Huber et al., 2021a) further extended this exploration to brain-wide mapping of the functional human connectome using the Shen268 atlas (Shen et al., 2013) which was also employed in our study. Several reported inter-regional layer-dependent FC patterns are consistent with our findings. However, the FOV in those studies restricts coverage of the temporal lobe, which is critical to memory processing and associated with disorders such as Alzheimer's disease. In contrast, our method provides broader brain coverage, making it more suitable for comprehensive layer-dependent FC mapping of the human connectome. This enhanced coverage allows the inclusion of regions critical for understanding large-scale brain networks and their roles in both health and disease.

Recent advancements in ultrahigh-resolution fMRI have enabled the exploration of brain-wide layer-specific FC across the human brain (see Table 1 for details). These imaging methods are broadly categorized as either BOLD-based or CBV-based. Here, we defined brain-wide acquisition as imaging

**Table 1.** List of brain-wide layer-dependent fMRI methods.
The imaging protocols need to cover more than half of the human brain to be classified as brain-wide acquisitions.

| Method | Contrast | TR | Coverage along z | Resolution | Field strength | Reference |
|---|---|---|---|---|---|---|
| VASO | CBV | 5.2 s for VASO +5.1 s for BOLD; 10.3 s effectively | 94.08 mm | 0.84 mm isotropic | 7T | Koiso et al., 2022 |
| MAGE VASO | CBV | 14.137 s (4 shots; 3.5 s per shot) | 100.8 mm | $0.82 \times 0.82 \times 0.84$ mm$^3$ | 3T | Huber et al., 2023 |
| EPIK | BOLD | 3.5 s | 108 mm | $0.51 \times 0.51 \times 1.0$ mm$^3$ | 7T | Yun et al., 2022 |
| Regular EPI | BOLD | 3 s | 55.5 mm | $0.85 \times 0.85 \times 1.5$ mm$^3$ | 7T | Deshpande et al., 2022 |
| 3D VAPER | CBV/CBF | 6 s per volume; 12 s effectively | 80.64 mm | $0.8 \times 0.8 \times 0.84$ mm$^3$ | 7T | Chai et al., 2024 |
| VN fMRI | BOLD | 4 s | 113.4 mm | $0.9 \times 0.9 \times 0.9$ mm$^3$ | 3T | This study |

protocols that cover more than half of the human brain, specifically >55 mm along the superior-inferior axis. Among the BOLD-based approaches, the EPI with keyhole (EPIK) method (*Yun et al., 2022*) enhances spatial resolution by fully sampling the central portion of k-space for every volume while sparsely sampling the outer portions. The EPIK method has demonstrated functional sensitivity by identifying several resting-state functional networks. Another BOLD-based technique has been employed to investigate layer-dependent cortico-thalamic FC and interhemispheric cortico-cortical FC (*Deshpande et al., 2022*). However, due to draining-vein contamination and lower resolution along the z-axis, the point spread function remains relatively flat, and the results require further experimental validation (*Deshpande et al., 2022*). For CBV-based methods, the spatial resolution achieves submillimeter precision along all three orthogonal axes (*Chai et al., 2024*; *Huber et al., 2023*; *Koiso et al., 2022*) with a maximum spatial coverage of 100.8 mm along the z-axis. However, the temporal resolution for CBV-based methods exceeds 5 s, raising concerns about temporal aliasing of resting-state signals. Although our method may not be as robust as CBV-based methods at the individual level, it successfully demonstrates layer specificity at the group level with a sample size of N=14.

## Limitations

Despite the unprecedented features of our VN fMRI, this study has some limitations. First, the VN fMRI method is inherently sensitive to T2*-related contamination. According to the results in *Figure 4*, stable activation in M1 was observed at the single-subject level across most scan protocols. Yet, the layer-dependent activation profiles in M1 were spatially unstable, irrespective of the application of VN gradients. This spatial instability is not entirely unexpected, as T2*-based contrast is inherently sensitive to various factors that perturb the magnetic field, such as eye movements, respiration, and macrovascular signal fluctuations. Furthermore, ICA-based artifact removal was intentionally omitted in *Figure 4* to ensure fair comparisons between protocols, leaving residual artifacts unaddressed. Inconsistency in performing the button-pressing task across sessions may also have contributed to the observed variability. These results suggest that submillimeter-resolution fMRI may not yet be suitable for reliable individual-level layer-dependent functional mapping, unless group-level statistics are incorporated to enhance robustness.

Additionally, the VN gradients may not sufficiently suppress the distal contributions of BOLD signals if the flow velocity in draining veins is relatively slow. Since penetrating veins are a potential source of cross-layer blurring, the flow velocity plays a critical role in the effectiveness of VN gradients. The flow velocity of draining veins is positively correlated with vessel diameter. For vessels with diameters exceeding 0.1 mm, flow velocities can exceed 15 mm/s (*Linninger et al., 2013*), which allows them to be largely suppressed by VN gradients. However, for vessels with diameters between 0.05 and 0.1 mm, flow velocities typically range from 5 to 15 mm/s, depending on vascular morphology. VN gradients may only partially suppress signals from veins within this size range. Although the signal contribution from individual vessels of this size may be negligible, the cumulative effect of multiple such vessels may become significant. Given that vessel density increases linearly from deep to superficial cortical layers (*Markuerkiaga et al., 2016*), residual signals may persist even after the application of VN gradients. Increasing the strength of VN gradients could mitigate this issue but might result in sensitivity loss.

Moreover, while our VN fMRI acquisition reduced the TR to 4.0 s, this duration remains suboptimal for studies utilizing event-related task designs. Future advancements in acceleration techniques would be highly beneficial for exploring high-level cognitive processes with greater temporal precision. However, such developments fall beyond the scope of this study.

## Conclusion

In summary, the developed VN fMRI exhibited layer specificity successfully at 3T. Leveraging its brain-wide coverage and reasonable scan time, the VN fMRI yielded promising results in the study of layer-specific FC. Given the widespread accessibility of 3T scanners, the potential impact of this development is expected to be extensive across various domains of neuroscience research.

# Materials and methods

## Participants

In this study, two cohorts of healthy adults were recruited. The first cohort consisted of four participants (3 males, age = 34.3 ± 9.2 years) for motor-task experiment. The second cohort was recruited for resting-state functional connectivity analysis and initially included 16 participants. Following the exclusion of two participants due to excessive head movement, the final sample for the second cohort comprised 14 participants (8 males, age = 30.0 ± 6.2 years). All participants underwent screening to confirm no history of neurological or psychiatric conditions, previous head trauma, or MRI contraindications. Only right-handed individuals are included in this study. Prior to participation, each individual provided informed consent in accordance with the experimental protocol approved by the University of North Carolina at Chapel Hill Institutional Review Board (IRB #19–2773).

## Stimulation paradigms

All visual stimuli across the paradigms were presented using PsychoPy software (Version 2022.2.4; *Brainard, 1997*). The stimuli were displayed on a screen positioned at the head end of the magnet bore, and participants viewed the visual presentations through a mirror mounted on the head coil.

### Button-pressing task

In a button-pressing task, the primary motor cortex (M1) receives incoming sensory and associative information in its superficial layers. This activation subsequently propagates to the deeper layers, where output signals are generated to ultimately control finger muscles. Consequently, both the superficial and deep layers of M1 exhibit activation, while the middle layers typically do not. Such a double-peak response pattern at M1 has been used as a hallmark to evaluate the spatial specificity in layer-dependent fMRI studies (*Chai et al., 2020*; *Huber et al., 2017a*; *Knudsen et al., 2023*; *Priovoulos et al., 2023*).

During the experiment, participants were given a button box and instructed to press the button with index and middle fingers of their right hand. The timing and frequency of the button pressing was synchronized with a video displayed on a screen within the scanner. The video consisted of a block-designed paradigm, as illustrated in *Figure 2a*. The button-pressing frequency was 2 Hz with each 'ON' block spanning 30 s, immediately followed by a 30-s rest during the 'OFF' block.

## Common protocol in MR acquisitions

MR images were acquired using a Siemens 3T Prisma scanner (Siemens Healthcare, Erlangen, Germany) and a 32-channel head coil at the Biomedical Research Imaging Center (BRIC) at the University of North Carolina at Chapel Hill. For each participant, an MPRAGE image was acquired for structural imaging using the following imaging parameters: 0.8 mm isotropic resolution, TR/TE/TI = 2400/2.24/1060ms, flip angle = 8°, in-plane acceleration factor = 2, partition thickness = 0.8 mm, 208 partitions, sagittal slicing, image matrix = 320 × 300, and FOV = 25.6 cm × 24.0 cm.

For functional imaging, the customized SMS sequence with VN gradient was implemented in the vendor-provided IDEA environment (VE11E). The functional data were acquired using a blipped-controlled aliasing in parallel imaging (blipped-CAIPI) SMS imaging (*Setsompop et al., 2012*) with two-dimensional (2D) single-shot EPI readout. While imaging protocols varied slightly across different experiments, they all shared the following parameters: isotropic spatial resolution of 0.9 mm; axial slicing; in-plane image dimensions of 224×210; frequency and phase encoding in the left-right and anterior-posterior directions, respectively; 126 slices; SMS factor of 3, and an in-plane acceleration rate of 3. Additionally, a brief fMRI acquisition with an opposite phase-encoding direction was acquired immediately before the functional sessions for distortion correction.

### Imaging protocols for task-based functional MRI

A button-pressing task was employed to assess BOLD sensitivity and spatial specificity. To identify the optimal b-value that effectively suppresses fast-flowing intravascular signals in large veins while minimizing signal reduction in localized capillaries and venules, empirical tests were performed using four parameter combinations: (1) b=0, TE = 30 ms; (2) b=0, TE = 38 ms; (3) b=6, TE = 38 ms; and (4)

b=8, TE = 39 ms. A total of 270 volumes were acquired, and the button-pressing task included 18 button-pressing blocks.

## Imaging protocols for resting-state functional MRI

Each participant in resting-state fMRI cohort underwent two resting-state fMRI sessions. Participants were instructed to stay motionless, keep their eyes closed, and avoid falling asleep. One of the sessions employed the VN gradient with a b value of 7, with the order of the two sessions being randomized among participants. For each session, a total of 300 volumes were acquired, taking approximately 21 min.

## Data preprocessing

The SMS-accelerated EPI time series were reconstructed using in-house MATLAB code, which performed slice GeneRalized Autocalibrating Partial Parallel Acquisition (slice-GRAPPA) (*Setsompop et al., 2012*) and in-plane GRAPPA (*Griswold et al., 2002*) jointly in one step. The multi-channel reconstructed images were subsequently combined using the adaptive combination method (*Walsh et al., 2000*). The reconstructed images were denoised using the NORDIC method with g-factor map (https://github.com/SteenMoeller/NORDIC_Raw, *Gau et al., 2025* function name: 'NIFTI_NORDIC'). The g-factor map was estimated from the image time series. The input images were complex-valued. The width of the smoothing filter for the phase was set as 10, while all other hyperparameters retained their default values. After NORDIC denoising, the time-series data were motion corrected using FSL, slice-timing corrected using sinc interpolation, and distortion corrected using FSL TOPUP (*Smith et al., 2004*). For artifact removal, we decomposed the magnitude image of the corrected data into a number of independent components by MELODIC (*Beckmann et al., 2005*). With the 28 resting-state runs acquired from 14 participants, we trained the ICA-based denoising tool FIX to auto-classify ICA components into signal and noise components (*Salimi-Khorshidi et al., 2014*). For resting-state fMRI datasets, auto-identified noise components were removed. For task-based fMRI datasets, however, artifact removal was not applied to ensure that only one experimental factor (b value or TE) differed between datasets, while all other preprocessing steps remained consistent. Finally, the time series were band-pass filtered between 0.01 and 0.1 Hz for resting-state fMRI and high-pass filtered at 0.01 Hz for task-based fMRI.

For task-based fMRI, additional coregistration was performed across sessions to enable comparisons between the four parameter combinations associated with the sessions. Filtered images from each session were temporally averaged and concatenated across sessions to create a unified dataset. The concatenated image series underwent motion correction using FSL's 'mcflirt' tool, resulting in the generation of a mean image across sessions. Subsequently, the filtered time series from all sessions were coregistered to this cross-session mean image using a linear transformation. This approach introduced interpolation errors across all the sessions, ensuring a fair and unbiased comparison.

## Phase regression

The phase regression approach was applied to remove the macrovascular component in the BOLD signal as indicated in previous reports (*Knudsen et al., 2023*; *Menon, 2002*). This macrovascular component not only contributes to alterations in signal magnitude but also introduces phase variations. By utilizing a linear regression approach between the time series of signal magnitude and phase, the macrovascular-associated contribution to the BOLD signal could be suppressed. Prior to phase regression, the time series of real and imaginary components were subjected to motion correction, followed by phase unwrapping. The phase regression was incorporated early in the data processing pipeline to minimize the discrepancy in data processing between magnitude and phase images (*Stanley et al., 2021*).

## Generation of a layered hand-knob label in M1

For depth-dependent analysis of the button-pressing task, the event-related responses were analyzed using FSL FEAT (*Woolrich et al., 2001*; *Woolrich et al., 2004*). Based on the BOLD activation maps, the hand knob region of the M1 was identified for each individual using the following criteria: (1) the hand knob region was required to be anatomically located in the precentral sulcus or gyrus; (2) it needed to exhibit consistent BOLD activation across the majority of testing conditions; and (3)

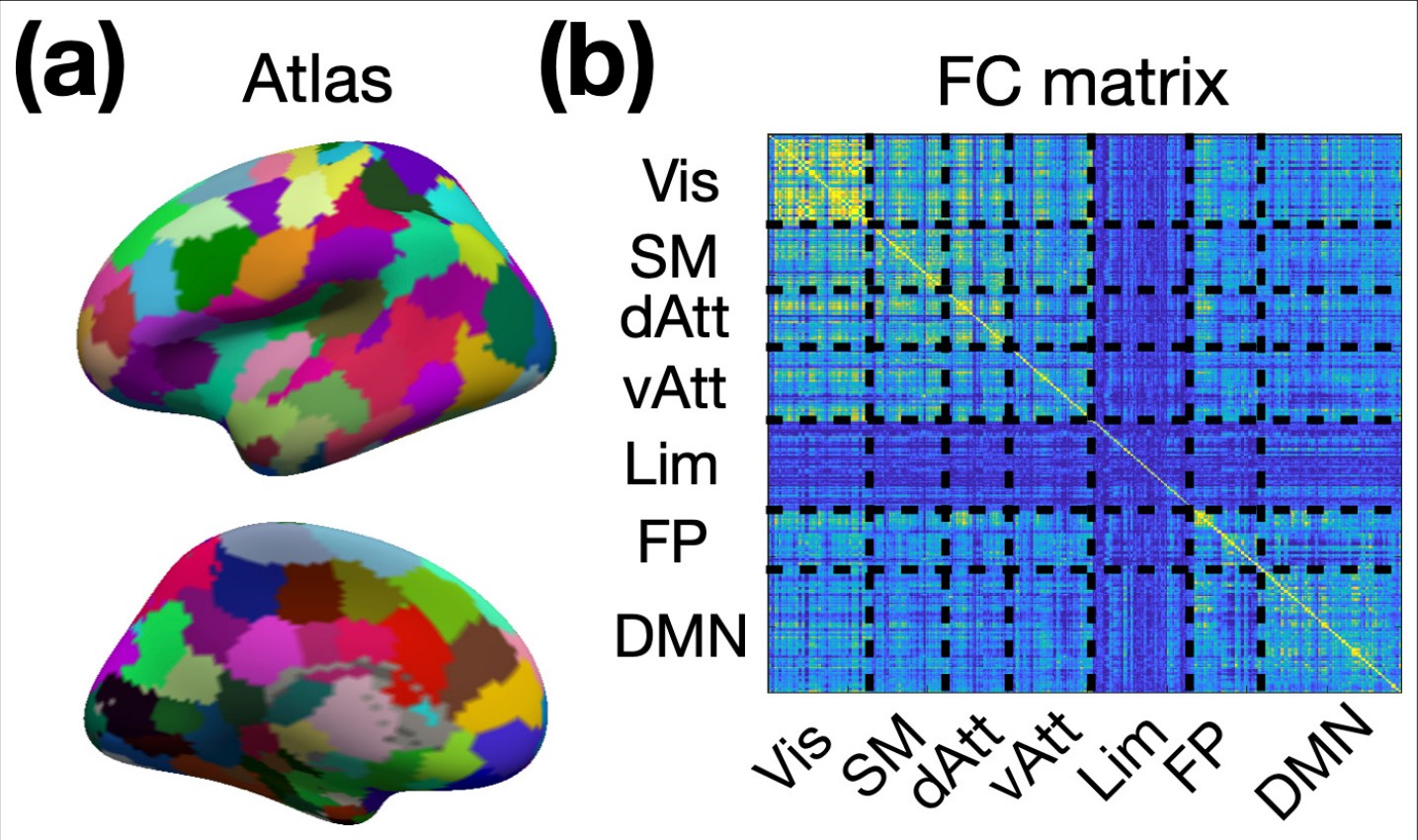

**Figure 10.** Generation of layer-specific functional connectivity. (**a**) The surface-based Shen268 functional parcellation. (**b**) The depth-dependent functional connectivity matrix. Abbreviations: Vis – visual network; SM – sensorimotor network; dAtt – dorsal attention network; vAtt – ventral attention network; Lim – limbic network; FP – frontoparietal network; DMN – default-mode network.

the region was expected to show BOLD activation in the deep cortical layers under the condition of b=0 and TE = 30 ms. Once the boundaries across cortical depth were defined, the gray matter boundaries of the hand knob region were delineated based on the T1-weighted anatomical image and the cortical ribbon mask but excluded the BOLD activation map to minimize potential bias in manual delineation.

The layer-dependent BOLD analysis involves the subsampling along cortical depth within the hand knob region. In this study, we generally follow the pipeline suggested by Dr. Huber (https://layerfmri. com/2018/03/11/quick-layering/#more-531). The image slice containing M1 was up-sampled by a factor of 5 using nearest-neighbor interpolation, achieving an in-plane resolution of 0.18 mm isotropic. With the created hand-knob label, we employed the LayNii software tool to form 20 equidistant layers (*Huber et al., 2021b*). The number 20 was determined according to the suggestion by LayNii that the number of layers may be set to at least four times larger than the resolution. Depth-dependent functional time series were extracted and averaged within each layer. The averaged layer-specific time series were then analyzed using a general linear model (GLM). The GLM utilized the double gamma function as the hemodynamic response function (HRF) (*Friston et al., 1998*) to model event-related responses.

## Resting-state functional connectivity with seeds at different layers

To assess inter-layer dependencies in resting-state data, seeds were generated in the superficial and deep layers of the primary motor cortex (M1) and primary sensory cortex (S1). Surface-based labels, obtained from FreeSurfer, were converted to volume-based labels using the mri_label2vol function. For the deep-layer seed, the projection fraction (proj) was set to 0.1, while for the superficial-layer seed, it was set to 0.9. The resulting volume-based label images were transformed from anatomical to functional image to enable FC analysis in native EPI space. Seed time courses were calculated by

averaging the signal across all voxels within each layer-specific label. These time courses were then used as regressors in a general linear model (GLM) implemented in FSL FEAT to generate FC maps.

To extract layer-specific FC values, the superficial, middle, and deep layers were resampled from EPI space onto individual cortical surfaces using mri_surf2vol with the following projection options: 'projfrac-avg 0.85 0.9 0.05', 'projfrac-avg 0.45 0.5 0.05', 'projfrac-avg 0.1 0.15 0.05' respectively. For overall FC values across cortical depth, the projection range 'projfrac-avg 0.1 0.9 0.05' was applied. The resulting surface-based FC maps were morphed from individual brain surfaces to a template brain using mri_surf2surf. Statistical significance of the group-averaged Fisher's z-transformed FC values was assessed using a t-test with the FreeSurfer command 'mri_glmfit'. Multiple comparison correction was performed using a Monte Carlo simulation implemented in FreeSurfer, with 4000 permutations. The vertex-wise and cluster-wise p-value thresholds were set to 0.05. The corrected p-values were then converted back to z-scores for better visualization.

## Surface-based layer-specific functional connectivity analysis

For surface-based subsampling along the cortical depth, we linearly coregistered the individual T1 volume onto the time-averaged EPI volume using Advanced Normalization Tools (ANTs; http://stnava. github.io/ANTs/). Following this, we resampled the processed EPI volumes onto the cortical surface at different cortical depths using the FreeSurfer command 'mri_vol2surf'. Adapting the layering strategy in volume space, we converted 20 cortical layers from volume to surface space. Here, the cortical depth ranged from –0.125 to 1.0625, with 0 and 1 representing the GM/WM and CSF/GM boundaries, respectively. Surface-based spatial smoothing was applied with full-width half-magnitude (FWHM) of 3 mm.

In order to investigate layer-specific functional connectivity throughout the brain, we employed the Shen268 functional parcellation (*Shen et al., 2013*). The volume-based Shen268 parcellation was converted onto the cortical surface as shown in *Figure 10a*. The volumetric EPI time series for each individual was resampled onto their individual brain surfaces before being projected onto the template brain surface. The template surface is of 'fsaverage5' from FreeSurfer. We also categorized the parcels in the Shen268 atlas into seven functional networks using the functional atlas reported by *Thomas Yeo et al., 2011*. The time course associated with each region-of-interest (ROI) was obtained by averaging all time courses within the ROI. Depth-dependent functional connectivity was calculated using Pearson correlation and then converted into Fisher's z values, as illustrated in *Figure 10b*. Group-level analyses were performed using one-sample t-tests on the Fisher's z-transformed correlation matrices.

## Acknowledgements

This work was supported in part by NIH grants R21AG060324.

## Additional information

### Funding

| Funder | Grant reference number | Author |
| --- | --- | --- |
| National Institutes of Health | R21AG060324 | Wei-Tang Chang |

The funders had no role in study design, data collection and interpretation, or the decision to submit the work for publication.

### Author contributions

Wei-Tang Chang, Conceptualization, Resources, Data curation, Software, Formal analysis, Supervision, Validation, Investigation, Visualization, Methodology, Writing – original draft, Project administration, Writing – review and editing; Weili Lin, Kelly S Giovanello, Resources

### Author ORCIDs

Wei-Tang Chang ⓘ https://orcid.org/0000-0002-2918-1752

## Ethics

This study was approved by the Institutional Review Board of the University of North Carolina at Chapel Hill (IRB #19-2773). All participants provided written informed consent prior to participation in accordance with institutional guidelines and the Declaration of Helsinki.

Reviewer #1 (Public review): https://doi.org/10.7554/eLife.92805.3.sa1
Reviewer #2 (Public review): https://doi.org/10.7554/eLife.92805.3.sa2
Reviewer #3 (Public review): https://doi.org/10.7554/eLife.92805.3.sa3
Author response https://doi.org/10.7554/eLife.92805.3.sa4

---

## Additional files

### Supplementary files
MDAR checklist

### Data availability

All imaging data has been deposited at OpenNeuro (https://openneuro.org/datasets/ds007543). Image reconstruction code is available at GitHub (copy archived at *welton0411, 2026*). Please note that the code provided is in its raw form; it has not been optimized, cleaned, or commented. While this may affect the ease of use or adaptation, we believe it remains a valuable resource for those interested in understanding or extending our analytical methods.

The following dataset was generated:

| Author(s) | Year | Dataset title | Dataset URL | Database and Identifier |
|---|---|---|---|---|
| Chang WT, Lin W, Giovanello K | 2026 | Open data of VNfMRI | https://openneuro.org/datasets/ds007543 | OpenNeuro, ds007543 |

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
