## [Editor Report · eLife Assessment]

This **useful** study presents a possible solution for a significant problem - that of draining vein sensitivity in functional MRI, which complicates the interpretability of laminar-fMRI results. The addition of a low diffusion-weighted gradient is presented to remove the draining vein signal and obtain functional responses with higher spatial fidelity. However, the strength of the evidence is **incomplete**, and most tests appear to have been done only in a single subject. Significance thresholds in presented maps are very low and most cortical depth-dependent response profiles do not differ from baseline, even in the BOLD data shown as reference. Curiously, even BOLD group data fails to replicate the well-known pattern of draining towards the cortical surface.

---

## [Referee Report · Reviewer #1 (Public review)]

Summary:

This study aims to provide imaging methods for users of the field of human layer-fMRI. This is an emerging field with 240 papers published so far. Different than implied in the manuscript, 3T is well represented among those papers. E.g. see the papers below that are not cited in the manuscript. Thus, the claim on the impact of developing 3T methodology for wider dissemination is not justified. Specifically, because some of the previous papers perform whole brain layer-fMRI (also at 3T) in more efficient, and more established procedures.

The authors implemented a sequence with lots of nice features. Including their own SMS EPI, diffusion bipolar pulses, eye-saturation bands, and they built their own reconstruction around it. This is not trivial. Only a few labs around the world have this level of engineering expertise. I applaud this technical achievement. However, I doubt that any of this is the right tool for layer-fMRI, nor does it represent an advancement for the field. In the thermal noise dominated regime of sub-millimeter fMRI (especially at 3T) it is established to use 3D readouts over 2D (SMS) readouts. While it is not trivial to implement SMS, the vendor implementations (as well as the CMRR and MGH implementations) are most widely applied across the majority of current fMRI studies already. The author's work on this does not serve any previous shortcomings in the field.

The mechanism to use bi-polar gradients to increase the localization specificity is doubtful to me. In my understanding, killing the intra-vascular BOLD should make it less specific. Also, the empirical data do not suggest a higher localization specificity to me.

Embedding this work in the literature of previous methods is incomplete. Recent trends of vessel signal manipulation with ABC or VAPER are not mentioned. Comparisons with VASO are outdated and incorrect.

The reproducibility of the methods and the result is doubtful (see below).

I don't think that this manuscript is in the top 50% of the 240 layer-fmri papers out there.

3T layer-fMRI papers that are not cited:

Taso, M., Munsch, F., Zhao, L., Alsop, D.C., 2021. Regional and depth-dependence of cortical blood-flow assessed with high-resolution Arterial Spin Labeling (ASL). Journal of Cerebral Blood Flow and Metabolism. https://doi.org/10.1177/0271678X20982382

Wu, P.Y., Chu, Y.H., Lin, J.F.L., Kuo, W.J., Lin, F.H., 2018. Feature-dependent intrinsic functional connectivity across cortical depths in the human auditory cortex. Scientific Reports 8, 1-14. https://doi.org/10.1038/s41598-018-31292-x

Lifshits, S., Tomer, O., Shamir, I., Barazany, D., Tsarfaty, G., Rosset, S., Assaf, Y., 2018. Resolution considerations in imaging of the cortical layers. NeuroImage 164, 112-120. https://doi.org/10.1016/j.neuroimage.2017.02.086

Puckett, A.M., Aquino, K.M., Robinson, P.A., Breakspear, M., Schira, M.M., 2016. The spatiotemporal hemodynamic response function for depth-dependent functional imaging of human cortex. NeuroImage 139, 240-248. https://doi.org/10.1016/j.neuroimage.2016.06.019

Olman, C.A., Inati, S., Heeger, D.J., 2007. The effect of large veins on spatial localization with GE BOLD at 3 T: Displacement, not blurring. NeuroImage 34, 1126-1135. https://doi.org/10.1016/j.neuroimage.2006.08.045

Ress, D., Glover, G.H., Liu, J., Wandell, B., 2007. Laminar profiles of functional activity in the human brain. NeuroImage 34, 74-84. https://doi.org/10.1016/j.neuroimage.2006.08.020

Huber, L., Kronbichler, L., Stirnberg, R., Ehses, P., Stocker, T., Fernández-Cabello, S., Poser, B.A., Kronbichler, M., 2023. Evaluating the capabilities and challenges of layer-fMRI VASO at 3T. Aperture Neuro 3. https://doi.org/10.52294/001c.85117

Scheeringa, R., Bonnefond, M., van Mourik, T., Jensen, O., Norris, D.G., Koopmans, P.J., 2022. Relating neural oscillations to laminar fMRI connectivity in visual cortex. Cerebral Cortex. https://doi.org/10.1093/cercor/bhac154

Strengths:

See above. The authors developed their own SMS sequence with many features. This is important to the field. And does not leave sequence development work to view isolated monopoly labs. This work democratises SMS.

The questions addressed here are of high relevance to the field: getting tools with good sensitivity, user-friendly applicability, and locally specific brain activity mapping is an important topic in the field of layer-fMRI.

Weaknesses:

(1) I feel the authors need to justify why flow-crushing helps localization specificity. There is an entire family of recent papers that aims to achieve higher localization specificity by doing the exact opposite. Namely, MT or ABC fRMRI aims to increase the localization specificity by highlighting the intravascular BOLD by means of suppressing non-flowing tissue. To name a few:

Priovoulos, N., de Oliveira, I.A.F., Poser, B.A., Norris, D.G., van der Zwaag, W., 2023. Combining arterial blood contrast with BOLD increases fMRI intracortical contrast. Human Brain Mapping hbm.26227. https://doi.org/10.1002/hbm.26227.

Pfaffenrot, V., Koopmans, P.J., 2022. Magnetization Transfer weighted laminar fMRI with multi-echo FLASH. NeuroImage 119725. https://doi.org/10.1016/j.neuroimage.2022.119725

Schulz, J., Fazal, Z., Metere, R., Marques, J.P., Norris, D.G., 2020. Arterial blood contrast (ABC) enabled by magnetization transfer (MT): a novel MRI technique for enhancing the measurement of brain activation changes. bioRxiv. https://doi.org/10.1101/2020.05.20.106666

Based on this literature, it seems that the proposed method will make the vein problem worse, not better. The authors could make it clearer how they reason that making GE-BOLD signals more extra-vascular weighted should help to reduce large vein effects.

The empirical evidence for the claim that flow crushing helps with the localization specificity should be made clearer. The response magnitude with and without flow crushing looks pretty much identical to me (see Fig, 6d).

It's unclear to me what to look for in Fig. 5. I cannot discern any layer patterns in these maps. It's too noisy. The two maps of TE=43ms look like identical copies from each other. Maybe an editorial error?

The authors discuss bipolar crushing with respect to SE-BOLD where it has been previously applied. For SE-BOLD at UHF, a substantial portion of the vein signal comes from the intravascular compartment. So I agree that for SE-BOLD, it makes sense to crush the intravascular signal. For GE-BOLD however, this reasoning does not hold. For GE-BOLD (even at 3T), most of the vein signal comes from extravascular dephasing around large unspecific veins and the bipolar crushing is not expected to help with this.

(2) The bipolar crushing is limited to one single direction of flow. This introduces a lot of artificial variance across the cortical folding pattern. This is not mentioned in the manuscript. There is an entire family of papers that perform layer-fmri with black-blood imaging that solves this with a 3D contrast preparation (VAPER) that is applied across a longer time period, thus killing the blood signal while it flows across all directions of the vascular tree. Here, the signal cruising is happening with a 2D readout as a "snap-shot" crushing. This does not allow the blood to flow in multiple directions.

VAPER also accounts for BOLD contaminations of larger draining veins by means of a tag-control sampling. The proposed approach here does not account for this contamination.

Chai, Y., Li, L., Huber, L., Poser, B.A., Bandettini, P.A., 2020. Integrated VASO and perfusion contrast: A new tool for laminar functional MRI. NeuroImage 207, 116358. https://doi.org/10.1016/j.neuroimage.2019.116358

Chai, Y., Liu, T.T., Marrett, S., Li, L., Khojandi, A., Handwerker, D.A., Alink, A., Muckli, L., Bandettini, P.A., 2021. Topographical and laminar distribution of audiovisual processing within human planum temporale. Progress in Neurobiology 102121. https://doi.org/10.1016/j.pneurobio.2021.102121

If I would recommend anyone to perform layer-fMRI with blood crushing, it seems that VAPER is the superior approach. The authors could make it clearer why users might want to use the unidirectional crushing instead.

(3) The comparison with VASO is misleading.

The authors claim that previous VASO approaches were limited by TRs of 8.2s. The authors might be advised to check the latest literature of the last years.

Koiso et al. has performed whole brain layer-fMRI VASO at 0.8mm at 3.9 seconds (with reliable activation) and 2.7 seconds (with unconvincing activation pattern, though), and 2.3 (without activation).

Also, whole brain layer-fMRI BOLD at 0.5mm and 0.7mm has been previously performed by the Juelich group at TRs of 3.5s (their TR definition is 'fishy' though).

Koiso, K., Müller, A.K., Akamatsu, K., Dresbach, S., Gulban, O.F., Goebel, R., Miyawaki, Y., Poser, B.A., Huber, L., 2023. Acquisition and processing methods of whole-brain layer-fMRI VASO and BOLD: The Kenshu dataset. Aperture Neuro 34. https://doi.org/10.1101/2022.08.19.504502

Yun, S.D., Pais‐Roldán, P., Palomero‐Gallagher, N., Shah, N.J., 2022. Mapping of whole‐cerebrum resting‐state networks using ultra‐high resolution acquisition protocols. Human Brain Mapping. https://doi.org/10.1002/hbm.25855

Pais-Roldan, P., Yun, S.D., Palomero-Gallagher, N., Shah, N.J., 2023. Cortical depth-dependent human fMRI of resting-state networks using EPIK. Front. Neurosci. 17, 1151544. https://doi.org/10.3389/fnins.2023.1151544

The authors are correct that VASO is not advised as a turn-key method for lower brain areas, incl. Hippocampus and subcortex. However, the authors use this word of caution that is intended for inexperienced "users" as a statement that this cannot be performed. This statement is taken out of context. This statement is not from the academic literature. It's advice for the 40+ user base that want to perform layer-fMRI as a plug-and-play routine tool in neuroscience usage. In fact, sub-millimeter VASO is routinely being performed by MRI-physicists across all brain areas (including deep brain structures, hippocampus etc). E.g. see Koiso et al. and an overview lecture from a layer-fMRI workshop that I had recently attended: https://youtu.be/kzh-nWXd54s?si=hoIJjLLIxFUJ4g20&t=2401

Thus, the authors could embed this phrasing into the context of their own method that they are proposing in the manuscript. E.g. the authors could state whether they think that their sequence has the potential to be disseminated across sites, considering that it requires slow offline reconstruction in Matlab?

Do the authors think that the results shown in Fig. 6c are suggesting turn-key acquisition of a routine mapping tool? In my humble opinion it looks like random noise, with most of the activation outside the ROI (in white matter).

(4) The repeatability of the results is questionable.

The authors perform experiments about the robustness of the method (line 620). The corresponding results are not suggesting any robustness to me. In fact the layer profiles in Fig. 4c vs. Fig 4d are completely opposite. Location of peaks turn into locations of dips and vice versa.

The methods are not described in enough detail to reproduce these results.

The authors mention that their image reconstruction is done "using in-house MATLAB code" (line 634). They do not post a link to github, nor do they say if they share this code.

It is not trivial to get good phase data for fMRI. The authors do not mention how they perform the respective coil-combination.

No data are shared for reproduction of the analysis.

(5) The application of NODRIC is not validated.

Previous applications of NORDIC at 3T layer-fMRI have resulted in mixed success. When not adjusted for the right SNR regime it can result in artifactual reductions of beta scores, depending on the SNR across layers. The authors could validate their application of NORDIC and confirm that the average layer-profiles are unaffected by the application of NORDIC. Also, the NORDIC version should be explicitly mentioned in the manuscript.

Akbari, A., Gati, J.S., Zeman, P., Liem, B., Menon, R.S., 2023. Layer Dependence of Monocular and Binocular Responses in Human Ocular Dominance Columns at 7T using VASO and BOLD (preprint). Neuroscience. https://doi.org/10.1101/2023.04.06.535924

Knudsen, L., Guo, F., Huang, J., Blicher, J.U., Lund, T.E., Zhou, Y., Zhang, P., Yang, Y., 2023. The laminar pattern of proprioceptive activation in human primary motor cortex. bioRxiv. https://doi.org/10.1101/2023.10.29.564658

Comments on revisions:

Among all the concerns mentioned above, I think there is only one of the specific issues that was sufficiently addressed.

The authors implemented a combination of three consecutive-dimensional flow crushers. Other concerns were not sufficiently addressed to change my confidence level of the study.

- While the abstract is still focusing on the utility of using 3T, they do not give credit to early 3T layer-fMRI papers leading the way to larger coverage and connectivity applications.

- While the author's choice of using custom SMS 2D readout is justified for them. I do not think that this very method will utilize widespread 3T whole brain connectivity experiments across the global 3T community. This lowers the impact of the paper.

- The images in Fig. 5 are still suspiciously similar. To the level that the noise pattern outside the brain is identical across large parts of the maps with and without PR.

- Maybe it's my ignorance, but I still do not agree why flow crushing focuses the local BOLD responses to small vessels.

- While my feel of a misleading representation of the literature had been accompanied by explicit references, the authors claim that they cannot find them?!? Or claim that they are about something else (which they are not, in my viewpoint).

Data and software are still not shared (not even example data, or nii data).

---

## [Referee Report · Reviewer #2 (Public review)]

This study developed a setup for laminar fMRI at 3T that aimed to get the best from all worlds in terms of brain coverage, temporal resolution, sensitivity to detect functional responses and spatial specificity. They used a gradient-echo EPI readout to facilitate sensitivity, brain coverage and temporal resolution. The former was additionally boosted by NORDIC denoising and the latter two were further supported by acceleration both in-plane and across slices. The authors evaluated whether the implementation of velocity-nulling (VN) gradients could mitigate macrovascular bias, known to hamper laminar specificity of gradient-echo BOLD.

Strengths:

The setup includes 0.9 mm isotropic acquisitions with large coverage at a reasonable TR. These parameters are hard to optimize simultaneously, and I applaud the ambitious attempt to get "the best from all worlds" (large coverage, high spatio/temporal resolution, spatial specificity, sensitivity), which is sought after in the field. Also, in terms of the availability of the method, it is favorable that it benefits from lower field strength (additional time for VN-gradient implementation, afforded by longer gray matter T2*). Furthermore, I like that the authors took steps to improve the original manuscript by e.g., collecting more data, adjusting the VN implementation to include flow-suppression along three rather than a single dimension, and adjusting the ROI-definition procedure to avoid circularity issues.

That being said, I still find the evidence weak in terms of this sequence achieving high spatial specificity and sensitivity. The results feel oversold and further validation is needed to make a case for the authors' conclusion that "[...] the potential impact of this development is expected to be extensive across various domains of neuroscience research". This is elaborated in the comments below:

The authors acknowledge that the VN setup in its current form probably does not suppress the impact of most ascending veins (these are also not targeted by phase regression, as most are probably too small to produce sufficiently large phase responses). This seems to limit the theoretical support for the author's claim of reduced inter-layer blurring (e.g. the claim that deep and superficial signals are less coupled with VN gradients than without based on Fig 6-7). This limitation withstanding, the method may still be helpful for limiting laminar dependencies by suppressing pial vein responses (which may carry signal from distant regions and layers that blur into superficial layers if left unsuppressed). Unfortunately, the empirical support of VN gradients suppressing superficial bias seems quite weak and is hard to evaluate. For example, the profiles in Figure 4 does not consistently show clearly less superficial bias when VN gradients are on - this might partly be due to the fact that clear bias was not always present in the profiles even without VN. I suspect this is largely explained by the selection of very small and quite unrepresentative ROIs. The corresponding activation maps appear strongly weighted towards CSF which is not always captured in the profile. I recommend sampling a much larger patch of cortex to more accurately capture the actual underlying bias. In this way, all non-VN profiles should have clear bias which should be clearly suppressed for VN if the method is effective. The authors do evaluate the effect of VN/phase regression based on a large activated region in visual cortex (Fig 5) - why not show laminar profiles from here, which is an obvious way to show the effect on superficial bias? I think such evaluations would be a more direct way of evaluating the methods impact on specificity, and are necessary for subsequent FC evaluations to be convincing.

The phase regression results are described inconsistently. In the results section, the authors, in my opinion, "correctly" acknowledge that phase regression seemed to have a very minor impact. However, in the discussion section it is described as if phase regression was effective in suppressing macrovascular responses (L 553-558), which the results do not support (especially based on profiles in Fig 4). There is barely any difference with/without phase regression, which may be due to the fact that ordinary least squares regression was chosen over a deming model which accounts for noise on the phase regressor. Although the authors correctly mentioned in their "answers to reviewers" that the required noise-ratio between magnitude and phase data can be hard to estimate, attempts of that has been described in previous phase regression studies which showed much larger effects (see e.g. Stanley et al. 2020, Knudsen et al. 2023).

I like that the authors put in additional efforts to provide analyses to validate their NORDIC implementation. However, this needs to be done on the VN setup directly, not the "regular BOLD setup" with b=0, since the ability of NORDIC to distinguish signal and noise components depends on CNR which is expected to deviate for these setups. Also, it seems z-scores and confidence intervals were computed based on GLM residuals which may lead to inflated z-values and overly narrow CI's due to reduced degrees of freedom following denoising. The denoised z-maps from Fig 3 indeed look somewhat strange, i.e. seemingly increased false positives (more salt/pepper and a bunch of white matter activation) with very weak hand knob activation. Also, something must be wrong with the CIs on the laminar profiles - they seem extremely narrow despite noise levels obviously being high for highly accelerated 3T submillimeter results extracted from a very small ROI. The authors may consider computing these statistics from variance across trials instead.

Given that the idea of the setup is to take advantage in terms of sensitivity by using GE-BOLD contrast relative to e.g. SE-EPI or CBV-weighted setups, they need to carefully demonstrate the sensitivity of their setup, which could be limited by high acceleration factors, the VN gradients, low field strength, etc. I like that they now put more emphasis on non-masked activation maps, but further comparison could be made through tSNR maps, raw single-volume images, raw timeseries, CNR based on across-trial variance, etc.

The major rationale for the setup is to achieve functional connectivity (FC) with brain-wide coverage at laminar resolutions, but it is framed as if this is something that has not been possible in the past with existing setups statements such as: "Despite advancements in acquisition speed, current CBV/CBF-based fMRI techniques remain inadequate for layer-dependent resting-state fMRI" (L138-140). To me, the functional connectivity results presented here with the VN setup are clearly less convincing than what has been shown with e.g. CBV-weighted acquisitions (e.g. Huber et al. 2021, Chai et al. 2024). The VN setup might also have advantages such as larger coverage as mentioned by the authors, but they fail to balance the comparison by highlighting where previous studies had clear edges. Thus, the impact of the results needs to be down-stated and a more balanced comparison with existing laminar FC studies is warranted. For example, acknowledging that the CBV-weighted studies demonstrate much higher spatial specificity.

Overall I would recommend a stronger emphasis on validating the claims about the sequence on task-based data for which there is a large body of literature to benchmark against (e.g. laminar fMRI studies in V1 and M1), before going to FC where the base for comparison and reference is much more limited in humans at laminar scales.

---

## [Referee Report · Reviewer #3 (Public review)]

Summary:

The authors are looking for a spatially specific functional brain response to visualise non-invasively with 3T (clinical field strength) MRI. They propose a velocity-nulled weighting to remove signal from draining veins in a submillimeter multiband acquisition.

Strengths:

- This manuscript addresses a real need in the cognitive neuroscience community interested in imaging responses in cortical layers in-vivo in humans.

- An additional benefit is the proposed implementation at 3T, a widely available field strength.

Weaknesses:

- The comparison in Figure 4 for different b-values shows % signal changes. However, as the baseline signal changes with added diffusion weighting, this is rather uninformative. A plot of t-values against cortical depth would be more insightful.

- Surprisingly, the %-signal change for a b-value of 0 is below 1% for 3/4 participants, even at the cortical surface. This raises some doubts about the task or ROI definition. A finger-tapping task should reliably engage the primary motor cortex, even at 3T, and even in individual participants.

- The double peak patter in the BOLD weighted images in Figure 4 is unexpected given the existing literature on BOLD responses as a function of cortical depth.

- Although I'd like to applaud the authors for their ambition with the connectivity analysis, the low significance threshold used in these maps (z=1,64) leads to concerns about the SNR of the underlying data.

I remain unconvinced of the conclusion that the developed VN fMRI exhibited layer specificity - the double peak which is taken as a marker of specificity is not absent in the BOLD responses either, and overall BOLD and VN response profiles as a function of cortical depth are quite similar.

---

## [Author Response]

The following is the authors’ response to the original reviews.

General responses:

The authors sincerely thank all the reviewers for their valuable and constructive comments. We also apologize for the long delay in providing this rebuttal due to logistical and funding challenges. In this revision, we modified the bipolar gradients from one single direction to all three directions. Additionally, in response to the concerns regarding data reliability, we conducted a thorough examination of each step in our data processing pipeline. In the original processing workflow, the projection-onto-convex-set (POCS) method was used for partial Fourier reconstruction. Upon examination, we found that applying the POCS method after parallel image reconstruction significantly altered the signal and resulted in considerable loss of functional feature. Futhermore, the original scan protocol employed a TE of 46 ms, which is notably longer than the typical TE of 33 ms. A prolonged TE can increase the ratio of extravascular to intravascular contributions. Importantly, the impact of TE on the efficacy of phase regression remains unclear, introducing potential confounding effects. To address these issues, we revised the protocol by shortening the TE from 46 ms to 39 ms. This adjustment was achieved by modifying the SMS factor to 3 and the in-plane acceleration rate to 3, thereby minimizing the confounding effects associated with an extended TE.

Following these changes, we recollected task-based fMRI data (N=4) and resting-state fMRI data (N=14) under the updated protocol. Using the revised dataset, we validated layer-specific functional connectivity (FC) through seed-based analyses. These analyses revealed distinct connectivity patterns in the superficial and deep layers of the primary motor cortex (M1), with statistically significant inter-layer differences. Furthermore, additional analyses with a seed in the primary sensory cortex (S1) corroborated the robustness and reliability of the revised methodology. We also changed the ‘directed’ functional connectivity in the title to ‘layer-specific’ functional connectivity, as drawing conclusions about directionality requires auxiliary evidence beyond the scope of this study.

We provide detailed responses to the reviewers’ comments below.

**Reviewer #1 (Public Review):**
Summary:(1) This study aims to provide imaging methods for users of the field of human layer-fMRI. This is an emerging field with 240 papers published so far. Different than implied in the manuscript, 3T is well represented among those papers. E.g. see the papers below that are not cited in the manuscript. Thus, the claim on the impact of developing 3T methodology for wider dissemination is not justified. Specifically, because some of the previous papers perform whole brain layer-fMRI (also at 3T) in more efficient, and more established procedures.3T layer-fMRI papers that are not cited:Taso, M., Munsch, F., Zhao, L., Alsop, D.C., 2021. Regional and depth-dependence of cortical blood-flow assessed with high-resolution Arterial Spin Labeling (ASL). Journal of Cerebral Blood Flow and Metabolism. https://doi.org/10.1177/0271678X20982382Wu, P.Y., Chu, Y.H., Lin, J.F.L., Kuo, W.J., Lin, F.H., 2018. Feature-dependent intrinsic functional connectivity across cortical depths in the human auditory cortex. Scientific Reports 8, 1-14. https://doi.org/10.1038/s41598-018-31292-xLifshits, S., Tomer, O., Shamir, I., Barazany, D., Tsarfaty, G., Rosset, S., Assaf, Y., 2018. Resolution considerations in imaging of the cortical layers. NeuroImage 164, 112-120. https://doi.org/10.1016/j.neuroimage.2017.02.086Puckett, A.M., Aquino, K.M., Robinson, P.A., Breakspear, M., Schira, M.M., 2016. The spatiotemporal hemodynamic response function for depth-dependent functional imaging of human cortex. NeuroImage 139, 240-248. https://doi.org/10.1016/j.neuroimage.2016.06.019Olman, C.A., Inati, S., Heeger, D.J., 2007. The effect of large veins on spatial localization with GE BOLD at 3 T: Displacement, not blurring. NeuroImage 34, 1126-1135. https://doi.org/10.1016/j.neuroimage.2006.08.045Ress, D., Glover, G.H., Liu, J., Wandell, B., 2007. Laminar profiles of functional activity in the human brain. NeuroImage 34, 74-84. https://doi.org/10.1016/j.neuroimage.2006.08.020Huber, L., Kronbichler, L., Stirnberg, R., Ehses, P., Stocker, T., Fernández-Cabello, S., Poser, B.A., Kronbichler, M., 2023. Evaluating the capabilities and challenges of layer-fMRI VASO at 3T. Aperture Neuro 3. https://doi.org/10.52294/001c.85117Scheeringa, R., Bonnefond, M., van Mourik, T., Jensen, O., Norris, D.G., Koopmans, P.J., 2022. Relating neural oscillations to laminar fMRI connectivity in visual cortex. Cerebral Cortex. https://doi.org/10.1093/cercor/bhac154

We thank the reviewer for listing out 8 papers related to 3T layer-fMRI papers. The primary goal of our work is to develop a methodology for brain-wide, layer-dependent resting-state functional connectivity at 3T. Upon review of the cited papers, we found that:

(1) One study (Lifshits et al.) was not an fMRI study.

(2) One study (Olman et al.) was conducted at 7T, not 3T.

(3) Two studies (Taso et al. and Wu et al.) employed relatively large voxel sizes (1.6 × 2.3 × 5 mm³ and 1.5 mm isotropic, respectively), which limits layer specificity.

(4) Only one of the listed studies (Huber et al., Aperture Neuro 2023) provides coverage of more than half of the brain.

While each of these studies offers valuable insights, the VASO study by Huber et al. is the most relevant to our work, given its brain-wide coverage. However, the VASO method employs a relatively long TR (14.137 s), which may not be optimal for resting-state functional connectivity analyses.

To address these limitations, our proposed method achieves submillimeter resolution, layer specificity, brain-wide coverage, and a significantly shorter TR (<5 s) altogether. We believe this advancement provides a meaningful contribution to the field, enabling broader applicability of layer-fMRI at 3T.

(2) The authors implemented a sequence with lots of nice features. Including their own SMS EPI, diffusion bipolar pulses, eye-saturation bands, and they built their own reconstruction around it. This is not trivial. Only a few labs around the world have this level of engineering expertise. I applaud this technical achievement. However, I doubt that any of this is the right tool for layer-fMRI, nor does it represent an advancement for the field. In the thermal noise dominated regime of sub-millimeter fMRI (especially at 3T), it is established to use 3D readouts over 2D (SMS) readouts. While it is not trivial to implement SMS, the vendor implementations (as well as the CMRR and MGH implementations) are most widely applied across the majority of current fMRI studies already. The author's work on this does not serve any previous shortcomings in the field.

We would like to thank the reviewer for their comments and the recognition of the technical efforts in implementing our sequence. We would like to address the points raised:

(1) We completely agree that in-house implementation of existing techniques does not constitute an advancement for the field. We did not claim otherwise in the manuscript. Our focus was on the development of a method for brain-wide, layer-dependent resting-state functional connectivity at 3T, as mentioned in the response above.

(2) The reviewer stated that "it is established to use 3D readouts over 2D (SMS) readouts". This is a strong claim, and we believe it requires robust evidence to support it. While it is true that 3D readouts can achieve higher tSNR in certain regions, such as the central brain, as shown in the study by Vizioli et al. (ISMRM 2020 abstract; https://cds.ismrm.org/protected/20MProceedings/PDFfiles/3825.html?utm_source=chatgpt.com), higher tSNR does not necessarily equate to improved detection power in fMRI studies. For instance, Le Ster et al. (PLOS ONE, 2019; https://doi.org/10.1371/journal.pone.0225286). demonstrated that while 3D EPI had higher tSNR in the central brain, SMS EPI produced higher t-scores in activation maps.

(3) When choosing between SMS EPI and 3D EPI, multiple factors should be taken into account, not just tSNR. For example, SMS EPI and 3D EPI differ in their sensitivity to motion and the complexity of motion correction. The choice between them depends on the specific research goals and practical constraints.

(4) We are open to different readout strategies, provided they can be demonstrated suitable to the research goals. In this study, we opted for 2D SMS primarily due to logistical considerations. This choice does not preclude the potential use of 3D readouts in the future if they are deemed more appropriate for the project objectives.

The mechanism to use bi-polar gradients to increase the localization specificity is doubtful to me. In my understanding, killing the intra-vascular BOLD should make it less specific. Also, the empirical data do not suggest a higher localization specificity to me.

We will elaborate the mechanism and reasoning in the later responses.

Embedding this work in the literature of previous methods is incomplete. Recent trends of vessel signal manipulation with ABC or VAPER are not mentioned. Comparisons with VASO are outdated and incorrect.The reproducibility of the methods and the result is doubtful (see below).

In this revision, we updated the scan protocol and recollected the imaging data. Detailed explanations and revised results are provided in the later responses.

I don't think that this manuscript is in the top 50% of the 240 layer-fmri papers out there.

We respect the reviewer’s personal opinion. However, we can only address scientific comments or critiques.

Strengths:See above. The authors developed their own SMS sequence with many features. This is important to the field. And does not leave sequence development work to view isolated monopoly labs. This work democratises SMS.The questions addressed here are of high relevance to the field: getting tools with good sensitivity, user-friendly applicability, and locally specific brain activity mapping is an important topic in the field of layer-fMRI.Weaknesses:(1) I feel the authors need to justify why flow-crushing helps localization specificity. There is an entire family of recent papers that aim to achieve higher localization specificity by doing the exact opposite. Namely, MT or ABC fRMRI aims to increase the localization specificity by highlighting the intravascular BOLD by means of suppressing non-flowing tissue. To name a few:Priovoulos, N., de Oliveira, I.A.F., Poser, B.A., Norris, D.G., van der Zwaag, W., 2023. Combining arterial blood contrast with BOLD increases fMRI intracortical contrast. Human Brain Mapping hbm.26227. https://doi.org/10.1002/hbm.26227.Pfaffenrot, V., Koopmans, P.J., 2022. Magnetization Transfer weighted laminar fMRI with multi-echo FLASH. NeuroImage 119725. https://doi.org/10.1016/j.neuroimage.2022.119725Schulz, J., Fazal, Z., Metere, R., Marques, J.P., Norris, D.G., 2020. Arterial blood contrast (ABC) enabled by magnetization transfer (MT): a novel MRI technique for enhancing the measurement of brain activation changes. bioRxiv. https://doi.org/10.1101/2020.05.20.106666Based on this literature, it seems that the proposed method will make the vein problem worse, not better. The authors could make it clearer how they reason that making GE-BOLD signals more extra-vascular weighted should help to reduce large vein effects.

The proposed VN fMRI method employs VN gradients to selectively suppress signals from fast-flowing blood in large vessels. Although this approach may initially appear to diverge from the principles of CBV-based techniques (Chai et al., 2020; Huber et al., 2017a; Pfaffenrot and Koopmans, 2022; Priovoulos et al., 2023), which enhance sensitivity to vascular changes in arterioles, capillaries, and venules while attenuating signals from static tissue and large veins, it aligns with the fundamental objective of all layer-specific fMRI methods. Specifically, these approaches aim to maximize spatial specificity by preserving signals proximal to neural activation sites and minimizing contributions from distal sources, irrespective of whether the signals are intra- or extra-vascular in origin. In the context of intravascular signals, CBV-based methods preferentially enhance sensitivity to functional changes in small vessels (proximal components) while demonstrating reduced sensitivity to functional changes in large vessels (distal components). For extravascular signals, functional changes are a mixture of proximal and distal influences. While tissue oxygenation near neural activation sites represents a proximal contribution, extravascular signal contamination from large pial veins reflects distal effects that are spatially remote from the site of neuronal activity. CBV-based techniques mitigate this challenge by unselectively suppressing signals from static tissues, thereby highlighting contributions from small vessels. In contrast, the VN fMRI method employs a targeted suppression strategy, selectively attenuating signals from large vessels (distal components) while preserving those from small vessels (proximal components). Furthermore, the use of a 3T scanner and the inclusion of phase regression in the VN approach mitigates contamination from large pial veins (distal components) while preserving signals reflecting local tissue oxygenation (proximal components). By integrating these mechanisms, VN fMRI improves spatial specificity, minimizing both intravascular and extravascular contributions that are distal to neuronal activation sites. We have incorporated the responses into Discussion section.

The empirical evidence for the claim that flow crushing helps with the localization specificity should be made clearer. The response magnitude with and without flow crushing looks pretty much identical to me (see Fig, 6d).

In the new results in Figure 4, the application of VN gradients attenuated the bias towards pial surface. Consistent with the results in Figure 4, Figure 5 also demonstrated the suppression of macrovascular signal by VN gradients.

It's unclear to me what to look for in Fig. 5. I cannot discern any layer patterns in these maps. It's too noisy. The two maps of TE=43ms look like identical copies from each other. Maybe an editorial error?

In this revision, the original Figure 5 has been removed. However, we would like to clarify that the two maps with TE = 43 ms in the original Figure 5 were not identical. This can be observed in the difference map provided in the right panel of the figure.

The authors discuss bipolar crushing with respect to SE-BOLD where it has been previously applied. For SE-BOLD at UHF, a substantial portion of the vein signal comes from the intravascular compartment. So I agree that for SE-BOLD, it makes sense to crush the intravascular signal. For GE-BOLD however, this reasoning does not hold. For GE-BOLD (even at 3T), most of the vein signal comes from extravascular dephasing around large unspecific veins, and the bipolar crushing is not expected to help with this.

The reviewer’s statement that "most of the vein signal comes from extravascular dephasing around large unspecific veins" may hold true for 7T. However, at 3T, the susceptibility-induced Larmor frequency shift is reduced by 57%, and the extravascular contribution decreases by more than 35%, as shown by Uludağ et al. 2009 (DOI: 10.1016/j.neuroimage.2009.05.051).

Additionally, according to the biophysical models (Ogawa et al., 1993; doi: 10.1016/S0006-3495(93)81441-3), the extravascular contamination from the pial surface is inversely proportional to the square of the distance from vessel. For a vessel diameter of 0.3 mm and an isotropic voxel size of 0.9 mm, the induced frequency shift is reduced by at least 36-fold at the next voxel. Notably, a vessel diameter of 0.3 mm is larger than most pial vessels. Theoretically, the extravascular effect contributes minimally to inter-layer dependency, particularly at 3T compared to 7T due to weaker susceptibility-related effects at lower field strengths. Empirically, as shown in Figure 7c, the results at M1 demonstrated that layer specificity can be achieved statistically with the application of VN gradients. We have incorporated this explanation into the Introduction and Discussion sections of the manuscript.

(2) The bipolar crushing is limited to one single direction of flow. This introduces a lot of artificial variance across the cortical folding pattern. This is not mentioned in the manuscript. There is an entire family of papers that perform layer-fmri with black-blood imaging that solves this with a 3D contrast preparation (VAPER) that is applied across a longer time period, thus killing the blood signal while it flows across all directions of the vascular tree. Here, the signal cruising is happening with a 2D readout as a "snap-shot" crushing. This does not allow the blood to flow in multiple directions.VAPER also accounts for BOLD contaminations of larger draining veins by means of a tag-control sampling. The proposed approach here does not account for this contamination.Chai, Y., Li, L., Huber, L., Poser, B.A., Bandettini, P.A., 2020. Integrated VASO and perfusion contrast: A new tool for laminar functional MRI. NeuroImage 207, 116358. https://doi.org/10.1016/j.neuroimage.2019.116358Chai, Y., Liu, T.T., Marrett, S., Li, L., Khojandi, A., Handwerker, D.A., Alink, A., Muckli, L., Bandettini, P.A., 2021. Topographical and laminar distribution of audiovisual processing within human planum temporale. Progress in Neurobiology 102121. https://doi.org/10.1016/j.pneurobio.2021.102121If I would recommend anyone to perform layer-fMRI with blood crushing, it seems that VAPER is the superior approach. The authors could make it clearer why users might want to use the unidirectional crushing instead.

We understand the reviewer’s concern regarding the directional limitation of bipolar crushing. As noted in the responses above, we have updated the bipolar gradient to include three orthogonal directions instead of a single direction. Furthermore, flow-related signal suppression does not necessarily require a longer time period. Bipolar diffusion gradients have been effectively used to nullify signals from fast-flowing blood, as demonstrated by Boxerman et al. (1995; DOI: 10.1002/mrm.1910340103). Their study showed that vessels with flow velocities producing phase changes greater than p radians due to bipolar gradients experience significant signal attenuation. The critical velocity for such attenuation can be calculated using the formula: 1/(2gGDd) where g is the gyromagnetic ratio, G is the gradient strength, d is the gradient pulse width and D is the time between the two bipolar gradient pulses. In the framework of Boxerman et al. at 1.5T, the critical velocity for b value of 10 s/mm^2^ is ~8 mm/s, resulting in a ~30% reduction in functional signal. In our 3T study, b values of 6, 7, and 8 s/mm^2^ correspond to critical velocities of 16.8, 15.2, and 13.9 mm/s, respectively. The flow velocities in capillaries and most venules remain well below these thresholds. Notably, in our VN fMRI sequences, bipolar gradients were applied in all three orthogonal directions, whereas in Boxerman et al.'s study, the gradients were applied only in the z-direction. Given the voxel dimensions of 3 × 3 × 7 mm^3^ in the 1.5T study, vessels within a large voxel are likely oriented in multiple directions, meaning that only a subset of fast-flowing signals would be attenuated. Therefore, our approach is expected to induce greater signal reduction, even at the same b values as those used in Boxerman et al.'s study. We have incorporated this text into the Discussion section of the manuscript.

(3) The comparison with VASO is misleading.The authors claim that previous VASO approaches were limited by TRs of 8.2s. The authors might be advised to check the latest literature of the last years.Koiso et al. performed whole brain layer-fMRI VASO at 0.8mm at 3.9 seconds (with reliable activation), 2.7 seconds (with unconvincing activation pattern, though), and 2.3 (without activation).Also, whole brain layer-fMRI BOLD at 0.5mm and 0.7mm has been previously performed by the Juelich group at TRs of 3.5s (their TR definition is 'fishy' though).Koiso, K., Müller, A.K., Akamatsu, K., Dresbach, S., Gulban, O.F., Goebel, R., Miyawaki, Y., Poser, B.A., Huber, L., 2023. Acquisition and processing methods of whole-brain layer-fMRI VASO and BOLD: The Kenshu dataset. Aperture Neuro 34. https://doi.org/10.1101/2022.08.19.504502Yun, S.D., Pais‐Roldán, P., Palomero‐Gallagher, N., Shah, N.J., 2022. Mapping of whole‐cerebrum resting‐state networks using ultra‐high resolution acquisition protocols. Human Brain Mapping. https://doi.org/10.1002/hbm.25855Pais-Roldan, P., Yun, S.D., Palomero-Gallagher, N., Shah, N.J., 2023. Cortical depth-dependent human fMRI of resting-state networks using EPIK. Front. Neurosci. 17, 1151544. https://doi.org/10.3389/fnins.2023.1151544

We thank the reviewer for providing these references. While the protocol with a TR of 3.9 seconds in Koiso’s work demonstrated reasonable activation patterns, it was not tested for layer specificity. Given that higher acceleration factors (AF) can cause spatial blurring, a protocol should only be eligible for comparison if layer specificity is demonstrated.

Secondly, the TRs reported in Koiso’s study pertain only to either the VASO or BOLD acquisition, not the combined CBV-based contrast. To generate CBV-based images, both VASO and BOLD data are required, effectively doubling the TR. For instance, if the protocol with a TR of 3.9 seconds is used, the effective TR becomes approximately 8 seconds. The stable protocol used by Koiso et al. to acquire whole-brain data (94.08 mm along the z-axis) required 5.2 seconds for VASO and 5.1 seconds for BOLD, resulting in an effective TR of 10.3 seconds. The spatial resolution achieved was 0.84 mm isotropic.

Unfortunately, we could not find the Juelich paper mentioned by the reviewer.

To have a more comprehensive comparison, we collated relevant literature on brain-wide layer-specific fMRI. We defined brain-wide acquisition as imaging protocols that cover more than half of the human brain, specifically exceeding 55 mm along the superior-inferior axis. We identified five studies and summarized their scan parameters, including effective TR, coverage, and spatial resolution, in Table 1.

The authors are correct that VASO is not advised as a turn-key method for lower brain areas, incl. Hippocampus and subcortex. However, the authors use this word of caution that is intended for inexperienced "users" as a statement that this cannot be performed. This statement is taken out of context. This statement is not from the academic literature. It's advice for the 40+ user base that wants to perform layer-fMRI as a plug-and-play routine tool in neuroscience usage. In fact, sub-millimeter VASO is routinely being performed by MRI-physicists across all brain areas (including deep brain structures, hippocampus etc). E.g. see Koiso et al. and an overview lecture from a layer-fMRI workshop that I had recently attended: https://youtu.be/kzh-nWXd54s?si=hoIJjLLIxFUJ4g20&t=2401

In this revision, we decided to focus on cortico-cortical functional connectivity and have removed the LGN-related content. Consequently, the text mentioned by the reviewer was also removed. Nevertheless, we apologize if our original description gave the impression that functional mapping of deep brain regions using VASO is not feasible. The word of caution we used is based on the layer-fMRI blog (https://layerfmri.com/2021/02/22/vaso_ve/) and reflects the challenges associated with this technique, as outlined by experts like Dr. Huber and Dr. Strinberg.

According to the information provided, including the video, functional mapping of the hippocampus and amygdala using VASO is indeed possible but remains technically challenging. The short arterial arrival times in these deep brain regions can complicate the acquisition, requiring RF inversion pulses to cover a wider area at the base of the brain. For example, as of 2023, four or more research groups were attempting to implement layer-fMRI VASO in the hippocampus. One such study at 3T required multiple inversion times to account for inflow effects, highlighting the technical complexity of these applications. This is the context in which we used the word of caution. We are not sure whether recent advancements like MAGEC VASO have improved its applicability. As of 2024, we have not identified any published VASO studies specifically targeting deep brain structures such as the hippocampus or amygdala. Therefore, it is difficult to conclude that “sub-millimeter VASO is routinely being performed by MRI physicists on deep brain structures such as the hippocampus.”

Thus, the authors could embed this phrasing into the context of their own method that they are proposing in the manuscript. E.g. the authors could state whether they think that their sequence has the potential to be disseminated across sites, considering that it requires slow offline reconstruction in Matlab?

We are enthusiastic about sharing our imaging sequence, provided its usefulness is conclusively established. However, it's important to note that without an online reconstruction capability, such as the ICE, the practical utility of the sequence may be limited. Unfortunately, we currently don’t have the manpower to implement the online reconstruction. Nevertheless, we are more than willing to share the offline reconstruction codes upon request.

Do the authors think that the results shown in Fig. 6c are suggesting turn-key acquisition of a routine mapping tool? In my humble opinion, it looks like random noise, with most of the activation outside the ROI (in white matter).

As we mentioned in the ‘general response’ in the beginning of the rebuttal, the POCS method for partial Fourier reconstruction caused the loss of functional feature, potentially accounting for the activation in white matter. In this revision, we have modified the pulse sequence, scan protocol and processing pipelines.

According to the results in Figure 4, stable activation in M1 was observed at the single-subject level across most scan protocols. Yet, the layer-dependent activation profiles in M1 were spatially unstable, irrespective of the application of VN gradients. This spatial instability is not entirely unexpected, as T2*-based contrast is inherently sensitive to various factors that perturb the magnetic field, such as eye movements, respiration, and macrovascular signal fluctuations. Furthermore, ICA-based artifact removal was intentionally omitted in Figure 4 to ensure fair comparisons between protocols, leaving residual artifacts unaddressed. Inconsistency in performing the button-pressing task across sessions may also have contributed to the observed variability. These results suggest that submillimeter-resolution fMRI may not yet be suitable for reliable individual-level layer-dependent functional mapping, unless group-level statistics are incorporated to enhance robustness. We have incorporated this text into the Limitation section of the manuscript.

(4) The repeatability of the results is questionable.The authors perform experiments about the robustness of the method (line 620). The corresponding results are not suggesting any robustness to me. In fact, the layer profiles in Fig. 4c vs. Fig 4d are completely opposite. The location of peaks turns into locations of dips and vice versa.The methods are not described in enough detail to reproduce these results.The authors mention that their image reconstruction is done "using in-house MATLAB code" (line 634). They do not post a link to github, nor do they say if they share this code.

We thank the reviewer for the comments regarding reproducibility and data sharing. In response, we have revised the Methods section and elaborated on the technical details to improve clarity and reproducibility.

Regarding code sharing, we acknowledge that the current in-house MATLAB reconstruction code requires further refinement to improve its readability and usability. Due to limited manpower, we have not yet been able to complete this task. However, we are committed to making the code publicly available and will upload it to GitHub as soon as the necessary resources are available.

For data sharing, we face logistical challenges due to the large size of the dataset, which spans tens of terabytes. Platforms like OpenNeuro, for example, typically support datasets up to 10TB, making it difficult to share the data in its entirety. Despite this limitation, we are more than willing to share offline reconstruction codes and raw data upon request to facilitate reproducibility.

Regarding data robustness, we kindly refer the reviewer to our response to the previous comment, where we addressed these concerns in greater detail.

It is not trivial to get good phase data for fMRI. The authors do not mention how they perform the respective coil-combination.No data are shared for reproduction of the analysis.

Obtaining phase data is relatively straightforward when the images are retrieved directly from raw data. For coil combination, we employed the adaptive coil combination approach described by (Walsh et al.; DOI: 10.1002/(sici)1522-2594(200005)43:5<682::aid-mrm10>3.0.co;2-g) The MATLAB code for this implementation was developed by Dr. Diego Hernando and is publicly available at https://github.com/welton0411/matlab .

(5) The application of NODRIC is not validated.Previous applications of NORDIC at 3T layer-fMRI have resulted in mixed success. When not adjusted for the right SNR regime it can result in artifactual reductions of beta scores, depending on the SNR across layers. The authors could validate their application of NORDIC and confirm that the average layer-profiles are unaffected by the application of NORDIC. Also, the NORDIC version should be explicitly mentioned in the manuscript.Akbari, A., Gati, J.S., Zeman, P., Liem, B., Menon, R.S., 2023. Layer Dependence of Monocular and Binocular Responses in Human Ocular Dominance Columns at 7T using VASO and BOLD (preprint). Neuroscience. https://doi.org/10.1101/2023.04.06.535924Knudsen, L., Guo, F., Huang, J., Blicher, J.U., Lund, T.E., Zhou, Y., Zhang, P., Yang, Y., 2023. The laminar pattern of proprioceptive activation in human primary motor cortex. bioRxiv. https://doi.org/10.1101/2023.10.29.564658

We appreciate the reviewer’s suggestion. To validate the application of NORDIC denoising in our study, we compared the BOLD activation maps before and after denoising in the visual and motor cortices, as well as the depth-dependent activation profiles in M1. These results are presented in Figure 3. The activation patterns in the denoised maps were consistent with those in the non-denoised maps but exhibited higher statistical significance. Notably, BOLD activation within M1 was only observed after NORDIC denoising, underscoring the necessity of this approach. Figure 3c shows the depth-dependent activation profiles in M1, highlighted by the green contours in Figure 3b. Both denoised and non-denoised profiles followed similar trends; however, as expected, the non-denoised profile exhibited larger confidence intervals compared to the NORDIC-denoised profile. These results confirm that NORDIC denoising enhances sensitivity without introducing distortions in the functional signal. The corresponding text has been incorporated into the Results section.

Regarding the implementation details of NORDIC denoising, the reconstructed images were denoised using a g-factor map (function name: NIFTI_NORDIC). The g-factor map was estimated from the image time series, and the input images were complex-valued. The width of the smoothing filter for the phase was set to 10, while all other hyperparameters were retained at their default values. This information has been integrated into the Methods section for clarity and reproducibility.

**Reviewer #2 (Public Review):**
This study developed a setup for laminar fMRI at 3T that aimed to get the best from all worlds in terms of brain coverage, temporal resolution, sensitivity to detect functional responses, and spatial specificity. They used a gradient-echo EPI readout to facilitate sensitivity, brain coverage and temporal resolution. The former was additionally boosted by NORDIC denoising and the latter two were further supported by parallel-imaging acceleration both in-plane and across slices. The authors evaluated whether the implementation of velocity-nulling (VN) gradients could mitigate macrovascular bias, known to hamper the laminar specificity of gradient-echo BOLD.The setup allows for 0.9 mm isotropic acquisitions with large coverage at a reasonable TR (at least for block designs) and the fMRI results presented here were acquired within practical scan-times of 12-18 minutes. Also, in terms of the availability of the method, it is favorable that it benefits from lower field strength (additional time for VN-gradient implementation, afforded by longer gray matter T2*).The well-known double peak feature in M1 during finger tapping was used as a test-bed to evaluate the spatial specificity. They were indeed able to demonstrate two distinct peaks in group-level laminar profiles extracted from M1 during finger tapping, which was largely free from superficial bias. This is rather intriguing as, even at 7T, clear peaks are usually only seen with spatially specific non-BOLD sequences. This is in line with their simple simulations, which nicely illustrated that, in theory, intravascular macrovascular signals should be suppressible with only minimal suppression of microvasculature when small b-values of the VN gradients are employed. However, the authors do not state how ROIs were defined making the validity of this finding unclear; were they defined from independent criteria or were they selected based on the region mostly expressing the double peak, which would clearly be circular? In any case, results are based on a very small sub-region of M1 in a single slice - it would be useful to see the generalizability of superficial-bias-free BOLD responses across a larger portion of M1.

We appreciate and understand the reviewer’s concerns. Given the small size of the hand knob region within M1 and its intersubject variability in location, defining this region automatically remains challenging. However, we applied specific criteria to minimize bias during the delineation of M1: (1) the hand knob region was required to be anatomically located in the precentral sulcus or gyrus; (2) it needed to exhibit consistent BOLD activation across the majority of testing conditions; and (3) the region was expected to show BOLD activation in the deep cortical layers under the condition of b = 0 and TE = 30 ms. Once the boundaries across cortical depth were defined, the gray matter boundaries of hand knob region were delineated based on the T1-weighted anatomical image and the cortical ribbon mask but excluded the BOLD activation map to minimize potential bias in manual delineation. Based on the new criteria, the resulting depth-dependent profiles, as shown in Figure 4, are no longer superficial-bias-free.

As repeatedly mentioned by the authors, a laminar fMRI setup must demonstrate adequate functional sensitivity to detect (in this case) BOLD responses. The sensitivity evaluation is unfortunately quite weak. It is mainly based on the argument that significant activation was found in a challenging sub-cortical region (LGN). However, it was a single participant, the activation map was not very convincing, and the demonstration of significant activation after considerable voxel-averaging is inadequate evidence to claim sufficient BOLD sensitivity. How well sensitivity is retained in the presence of VN gradients, high acceleration factors, etc., is therefore unclear. The ability of the setup to obtain meaningful functional connectivity results is reassuring, yet, more elaborate comparison with e.g., the conventional BOLD setup (no VN gradients) is warranted, for example by comparison of tSNR, quantification and comparison of CNR, illustration of unmasked-full-slice activation maps to compare noise-levels, comparison of the across-trial variance in each subject, etc. Furthermore, as NORDIC appears to be a cornerstone to enable submillimeter resolution in this setup at 3T, it is critical to evaluate its impact on the data through comparison with non-denoised data, which is currently lacking.

We appreciate the reviewer’s comments and acknowledge that the LGN results from a single participant were not sufficiently convincing. In this revision, we have removed the LGN-related results and focused on cortico-cortical FC. To evaluate data quality, we opted to present BOLD activation maps rather than tSNR, as high tSNR does not necessarily translate to high functional significance. In Figure 3, we illustrate the effect of NORDIC denoising, including activation maps and depth-dependent profiles. Figure 4 presents activation maps acquired under different TE and b values, demonstrating that VN gradients effectively reduce the bias toward the pial surface without altering the overall activation patterns. The results in Figure 4 and Figure 5 provide evidence that VN gradients retain sensitivity while reducing superficial bias. The ability of the setup to obtain meaningful FC results was validated through seed-based analyses, identifying distinct connectivity patterns in the superficial and deep layers of the primary motor cortex (M1), with significant inter-layer differences (see Figure 7). Further analyses with a seed in the primary sensory cortex (S1) demonstrated the reliability of the method (see Figure 8). For further details on the results, including the impact of VN gradients and NORDIC denoising, please refer to Figures 3 to 8 in the Results section.

Additionally, we acknowledge the limitations of our current protocol for submillimeter-resolution fMRI at the individual level. We found that robust layer-dependent functional mapping often requires group-level statistics to enhance reliability. This issue has been discussed in detail in the Limitations section.

The proposed setup might potentially be valuable to the field, which is continuously searching for techniques to achieve laminar specificity in gradient echo EPI acquisitions. Nonetheless, the above considerations need to be tackled to make a convincing case.

**Reviewer #3 (Public Review):**
Summary:The authors are looking for a spatially specific functional brain response to visualise non-invasively with 3T (clinical field strength) MRI. They propose a velocity-nulled weighting to remove the signal from draining veins in a submillimeter multiband acquisition.Strengths:- This manuscript addresses a real need in the cognitive neuroscience community interested in imaging responses in cortical layers in-vivo in humans.- An additional benefit is the proposed implementation at 3T, a widely available field strength.Weaknesses:- Although the VASO acquisition is discussed in the introduction section, the VN-sequence seems closer to diffusion-weighted functional MRI. The authors should make it more clear to the reader what the differences are, and how results are expected to differ. Generally, it is not so clear why the introduction is so focused on the VASO acquisition (which, curiously, lacks a reference to Lu et al 2013). There are many more alternatives to BOLD-weighted imaging for fMRI. CBF-weighted ASL and GRASE have been around for a while, ABC and double-SE have been proposed more recently.

The major distinction between diffusion-weighted fMRI (DW-fMRI) and our methodology lies in the b-value employed. DW-fMRI typically measures cellular swelling using b-values greater than 1000 s/mm^2^ e.g., 1800 s/mm(sup>2). In contrast, our VN-fMRI approach measures hemodynamic responses by employing smaller b-values specifically designed to suppress signals from fast-flowing draining veins rather than detecting microstructural changes.

Regarding other functional contrasts, we agree that more layer-dependent fMRI approaches should be mentioned. In this revision, we have expanded the Introduction section to include discussions of the double spin-echo approach and CBV-based methods, such as MT-weighted fMRI, VAPER, ABC, and CBF-based method ASL. Additionally, the reference to Lu et al. (2013) has been cited in the revised manuscript. The corresponding text has been incorporated into the Introduction section to provide a more comprehensive overview of alternative functional imaging techniques.

- The comparison in Figure 2 for different b-values shows % signal changes. However, as the baseline signal changes dramatically with added diffusion weighting, this is rather uninformative. A plot of t-values against cortical depth would be much more insightful.- Surprisingly, the %-signal change for a b-value of 0 is not significantly different from 0 in the gray matter. This raises some doubts about the task or ROI definition. A finger-tapping task should reliably engage the primary motor cortex, even at 3T, and even in a single participant.- The BOLD weighted images in Figure 3 show a very clear double-peak pattern. This contradicts the results in Figure 2 and is unexpected given the existing literature on BOLD responses as a function of cortical depth.- Given that data from Figures 2, 3, and 4 are derived from a single participant each, order and attention affects might have dramatically affected the observed patterns. Especially for Figure 4, neither BOLD nor VN profiles are really different from 0, and without statistical values or inter-subject averaging, these cannot be used to draw conclusions from.

We appreciate the reviewer’s suggestions. In this revision, we have made significant updates to the participant recruitment, scan protocol, data processing, and M1 delineation. Please refer to the "General Responses" at the beginning of the rebuttal and the first response to Reviewer #2 for more details.

Previously, the variation in depth-dependent profiles was calculated across upscaled voxels within a specific layer. However, due to the small size of the hand knob region, the number of within-layer voxels was limited, resulting in inaccurate estimations of signal variation. In the revised manuscript, the signal was averaged within each layer before performing the GLM analysis, and signal variation was calculated using the temporal residuals. The technical details of these changes are described in the "Materials and Methods" section. Furthermore, while the initial submission used percentage signal change for the profiles of M1, the dramatic baseline fluctuations observed previously are no longer an issue after the modifications. For this reason, we retained the use of percentage signal change to present the depth-dependent profiles. After these adjustments, the profiles exhibited a bias toward the pial surface, particularly in the absence of VN gradients.

- In Figure 5, a phase regression is added to the data presented in Figure 4. However, for a phase regression to work, there has to be a (macrovascular) response to start with. As none of the responses in Figure 4 are significant for the single participant dataset, phase regression should probably not have been undertaken. In this case, the functional 'responses' appear to increase with phase regression, which is contra-intuitive and deserves an explanation.

We agreed with reviewer’s argument. In the revised results, the issues mentioned by the reviewer are largely diminished. The updated analyses demonstrate that phase regression effectively reduces superficial bias, as shown in Figures 4 and 5.

- Consistency of responses is indeed expected to increase by a removal of the more variable vascular component. However, the microvascular component is always expected to be smaller than the combination of microvascular + macrovascular responses. Note that the use of %signal changes may obscure this effect somewhat because of the modified baseline. Another expected feature of BOLD profiles containing both micro- and microvasculature is the draining towards the cortical surface. In the profiles shown in Figure 7, this is completely absent. In the group data, no significant responses to the task are shown anywhere in the cortical ribbon.

We agreed with reviewer’s comments. In the revised manuscript, the results have been substantially updated to addressing the concerns raised. The original Figure 7 is no longer relevant and has been removed.

- Although I'd like to applaud the authors for their ambition with the connectivity analysis, I feel that acquisitions that are so SNR starved as to fail to show a significant response to a motor task should not be used for brain wide directed connectivity analysis.

We appreciate the reviewer’s comments and share the concern about SNR limitations. In the updated results presented in Figure 5, the activation patterns in the visual cortex were consistent across TEs and b values. At the motor cortex, stable activation in M1 was observed at the single-subject level across most scan protocols. However, the layer-dependent activation profiles in M1 exhibited spatial instability, irrespective of the application of VN gradients. This spatial instability is not entirely unexpected, as T2*-based contrast is inherently sensitive to factors that perturb the magnetic field, such as eye movements, respiration, and macrovascular signal fluctuations. Additionally, ICA-based artifact removal was intentionally omitted in Figure 4 to ensure fair comparisons across protocols, leaving some residual artifacts unaddressed. Variability in task performance during button-pressing sessions may have further contributed to the observed inconsistencies.

Although these findings suggest that submillimeter-resolution fMRI may not yet be reliable for individual-level layer-dependent functional mapping, the group-level FC analyses can still yield robust results. In Figure 7, group-level statistics revealed distinct functional connectivity (FC) patterns associated with superficial and deep layers in M1. These FC maps exhibited significant differences between layers, demonstrating that VN fMRI enhances inter-layer independence. Additional FC analyses with a seed placed in S1 further validated these findings (see Figure 8).

The claim of specificity is supported by the observation of the double-peak pattern in the motor cortex, previously shown in multiple non-BOLD studies. However, this same pattern is shown in some of the BOLD weighted data, which seems to suggest that the double-peak pattern is not solely due to the added velocity nulling gradients. In addition, the well-known draining towards the cortical surface is not replicated for the BOLD-weighted data in Figures 3, 4, or 7. This puts some doubt about the data actually having the SNR to draw conclusions about the observed patterns.

We appreciate the reviewer’s comments. In the updated results, the efficacy of the VN gradients is evident near the pial surface, as shown in Figures 4 and 5. In Figure 4, comparing the second and third columns (b = 0 and b = 6 s/mm^2^, respectively, at TE = 38 ms), the percentage signal change in the superficial layers is generally lower with b = 6 s/mm^2^ than with b = 0. This indicates that VN gradient-induced signal suppression is more pronounced in the superficial layers. Additionally, in Figure 5, the VN gradients effectively suppressed macrovascular signals as highlighted by the blue circles. These observations support the role of VN gradients in enhancing specificity by reducing superficial bias and macrovascular contamination. Furthermore, bias towards cortical surface was observed in the updated results in Figure 4.

**Recommendations for the authors:**
Reviewer #2 (Recommendations For The Authors):(1) L141: "depth dependent" is slightly misleading here. It could be misunderstood to suggest that the authors are assessing how spatial specificity varies as a function of depth. Rather, they are assessing spatial specificity based on depth-dependent responses (double peak feature). Perhaps "layer-dependent spatial specificity" could be substituted with laminar specificity?

We thank the reviewer for the suggestion. The term “depth dependent” has been replaced by “layer dependent” in the revised manuscript.

(2) L146-149: these do not validate spatial specificity.

The original text is removed.

(3) L180: Maybe helpful to describe what the b-value is to assist unfamiliar readers.

We have clarified the b-value as “the strength of the bipolar diffusion gradients” where it is first mentioned in the manuscript.

(4) Figure 1B: I think it would be appropriate with a sentence of how the authors define micro/macrovasculature. Figure 1B seems to suggest that large ascending veins are considered microvascular which I believe is a bit unconventional. Nevertheless, as long as it is clearly stated, it should be fine.

In our context, macrovasculature refers to vessels that are distal to neural activation sites and contribute to extravascular contamination. These vessels are typically larger in size (e.g., > 0.1 mm in diameter) and exhibit faster flow rates (e.g., > 10 mm/s).

(5) I think the authors could be more upfront with the point about non-suppressed extravascular effects from macrovasculature, which was briefly mentioned in the discussion. It could already be highlighted in the introduction or theory section.

We thank the reviewer’s suggestions. We have expanded the discussion of extravascular effects from macrovasculature in both the Introduction (5th paragraph) and Discussion (3rd paragraph) sections.

(6) The phase regression figure feels a bit misplaced to me. If the authors agree: rather than showing the TE-dependency of the effect of phase regression, it may be more relevant for the present study to compare the conventional setup with phase regression, with the VN setup without phase regression. I.e., to show how the proposed setup compares to existing 3T laminar fMRI studies.

In this revision, both the TE-dependent and VN-dependent effects of phase regression were investigated. The results in Figure 4 and Figure 5 demonstrated that phase regression effectively suppresses macrovascular contributions primarily near the gray matter/CSF boundary, irrespective of TE or the presence of VN gradients.

(7) L520: It might be beneficial to also cite the large body of other laminar studies showing the double peak feature to underscore that it is highly robust, which increases its relevance as a test-bed to assess spatial specificity.

We agreed. More literatures have been cited (Chai et al., 2020; Huber et al., 2017a; Knudsen et al., 2023; Priovoulos et al., 2023).

(8) L557: The argument that only one participant was assessed to reduce inter-subject variability is hard to buy. If significant variability exists across subjects, this would be highly relevant to the authors and something they would want to capture.

We thank the reviewer for the suggestions. In this revision, we have increased the number of participants to 4 for protocol development and 14 for resting-state functional connectivity analysis, allowing us to better assess and account for inter-subject variability.

(9) L637: add download link and version number.

The download link has been added as requested. The version number is not applicable.

(10) L638: How was the phase data coil-combined?

The reconstructed multi-channel data, which were of complex values, were combined using the adaptive combination method (Walsh et al.; DOI: 10.1002/(sici)1522-2594(200005)43:5<682::aid-mrm10>3.0.co;2-g). The MATLAB code for this implementation was developed by Dr. Diego Hernando and is publicly available at https://github.com/welton0411/matlab . The phase data were then extracted using the MATLAB function ‘angle’.

(11) L639: Why was the smoothing filter parameter changed (other parameters were default)?

The smoothing filter parameter was set based on the suggestion provided in the help comments of the NIFTI_NORDIC function:

function NIFTI_NORDIC(fn_magn_in,fn_phase_in,fn_out,ARG)

% fMRI

%

% ARG.phase_filter_width=10;

In other words, we simply followed the recommendation outlined in the NIFTI_NORDIC function’s documentation.

(12) I assume the phase data was motion corrected after transforming to real and imaginary components and using parameters estimated from magnitude data? Maybe add a few sentences about this.

Prior to phase regression, the time series of real and imaginary components were subjected to motion correction, followed by phase unwrapping. The phase regression was incorporated early in the data processing pipeline to minimize the discrepancy in data processing between magnitude and phase images (Stanley et al., 2021).

(13) Was phase regression applied with e.g., a deming model, which accounts for noise on both the x and y variable? In my experience, this makes a huge difference compared with regular OLS.

We appreciate the reviewer’s insightful comment. We are aware that the noise present in both magnitude and phase data therefore linear Deming regression would be a good fit to phase regression (Stanley et al., 2021). To perform Deming regression, however, the ratio of magnitude error variance to phase error variance must be predefined. In our initial tests, we found that the regression results were sensitive to this ratio. To avoid potential confounding, we opted to use OLS regression for the current analysis. However, we agreed Deming model could enhance the efficacy of phase regression if the ratio could be determined objectively and properly.

(14) Figure 2: What is error bar reflecting? I don't think the across-voxel error, as also used in Figure 4, is super meaningful as it assumes the same response of all voxels within a layer (might be alright for such a small ROI). Would it be better to e.g. estimate single-trial response magnitude (percent signal change) and assess variability across? Also, it is not obvious to me why b=30 was chosen. The authors argue that larger values may kill signal, but based on this Figure in isolation, b=48 did not have smaller response magnitudes (larger if anything).

We agreed with the reviewer’s opinion on the across-voxel error. In the revised manuscript, the signal was averaged within each layer before performing the GLM analysis, and signal variation was calculated using the temporal residuals. The technical details of these changes are described in the "Materials and Methods" section.

Additionally, the bipolar diffusion gradients were modified from a single direction to three orthogonal directions. As a result, the questions and results related to b=30 or b=48 are no longer applicable.

(15) Figure 5: would be informative to quantify the effect of phase regression over a large ROI and evaluate reduction in macrovascular influence from superficial bias in laminar profiles.

We appreciate the reviewer’s suggestion. In the revised manuscript, the reduction in macrovascular influence from superficial bias across a large ROI is displayed in Figure 5. Additionally, the impact on laminar profiles is demonstrated in Figure 4.

(16) L406-408: What kind of robustness?

We acknowledge that describing the protocol as “robust” was an overstatement. The updated results indicate that the current protocol for submillimeter fMRI may not yet be suitable for reliable individual-level layer-dependent functional mapping. However, group-level functional connectivity (FC) analyses demonstrated clear layer-specific distinctions with VN fMRI, which were not evident in conventional fMRI. These findings highlight the enhanced layer specificity achievable with VN fMRI.

(17) Figure 8: I think (C) needs pointers to superficial, middle, and deep layers? Why is it not in the same format as in Figure 9C? The discussion of the FC results could benefit from more references supporting that these observations are in line with the literature.

In the revised results, the layer pooling shown in Figure 9c has been removed, making the question regarding format alignment no longer applicable. Additionally, references supporting the FC results have been added to the revised Discussion section (7th paragraph).

(18) L456-457: But correlation coefficients may also be biased by different CNR across layers.

That is correct. In the updated FC results in Figure 7 to 9, we used group-level statistics rather than correlation coefficients.

**Reviewer #3 (Recommendations For The Authors):**
The results in Figure 2-6 should be repeated over, or averaged over, a (small) group of participants. N=6 is usual in this field. I would seriously reconsider the multiband acceleration - the acquisition seemingly cannot support the SNR hit.A few more specific points are given below:(1) Abstract: The sentence about LGN in the abstract came for me out of the blue - why would LGN be important here, it's not even a motor network node? Perhaps the aims of the study should be made more clear - if it's about networks as suggested earlier then a network analysis result would be expected too. Expanding the directed FC findings would improve the logical flow of the abstract. Given the many concerns, removing the connectivity analysis altogether would also be an option.

We thank the reviewer for the suggestions. The LGN-related results indeed diluted the focus of this study and have been completely removed in this revision.

(2) Line 105: in addition to the VASO method, ..

The corresponding text has been revised, and as a result, the reviewer’s suggestion is no longer applicable.

(3) If out of the set MB 4 / 5 / 6 MB4 was best, why did the authors not continue with a comparison including MB3 and MB2? It seems to me unlikely that the MB4 acquisition is actually optimal.

Results: We appreciate the reviewer’s suggestions. In this revision, we decreased the MB factor to 3, as it allowed us to increase the in-plane acceleration rate to 3, thereby shortening the TE. The resulting sensitivity for both individual and group-level results is detailed in earlier responses, such as the response to Q16 for Reviewer #2.

(4) The formatting of the references is occasionally flawed, including first names and/or initials. Please consider using a reliable reference manager.

We used Zotero as our reference manager in this revision to ensure consistency and accuracy. The references have been formatted according to the APA style.

(5) In the caption of Figure 5, corrected and uncorrected p values are identical. What multiple comparisons correction was made here? A multiple comparisions over voxels (as is standard) would usually lead to a cut-off ~z=3.2. That would remove most of the 'responses' shown in figure 5.

We appreciate the reviewer’s comment. The original results presented in Figure 5 have been removed in the revised manuscript, making this comment no longer applicable.